# Obligate mutualistic cooperation limits evolvability

Benedikt Pauli [1], Leonardo Oña [1], Marita Hermann [1,2] & Christian Kost [1✉]

Cooperative mutualisms are widespread and play fundamental roles in many ecosystems. Given that these interactions are often obligate, the Darwinian fitness of the participating individuals is not only determined by the information encoded in their own genomes, but also the traits and capabilities of their corresponding interaction partners. Thus, a major outstanding question is how obligate cooperative mutualisms affect the ability of organisms to adapt evolutionarily to changing environmental conditions. Here we address this issue using a mutualistic cooperation between two auxotrophic genotypes of *Escherichia coli* that reciprocally exchanged costly amino acids. Amino acid-supplemented monocultures and unsupplemented cocultures were exposed to stepwise increasing concentrations of different antibiotics. This selection experiment reveals that metabolically interdependent bacteria are generally less able to adapt to environmental stress than autonomously growing strains. Moreover, obligate cooperative mutualists frequently regain metabolic autonomy, resulting in a collapse of the mutualistic interaction. Together, our results identify a limited evolvability as a significant evolutionary cost that individuals have to pay when entering into an obligate mutualistic cooperation.

[1] Department of Ecology, Osnabrück University, Barbarastraße 13, 49076 Osnabrück, Germany. [2]Present address: Department of Plant Physiology, Osnabrück University, Barbarastr. 11, 49076 Osnabrück, Germany. ✉email: christiankost@gmail.com

Mutualistic interactions have been key for the evolution of life on earth[1]. By cooperating with members of the same or a different species, organisms gain novel capabilities without having to autonomously evolve these traits[2–4]. If the benefits resulting from these reciprocal interactions strongly outweigh the costs, significant fitness advantages can result for the interacting individuals, relative to organisms not engaging in such an interaction[5,6]. However, an almost inevitable consequence of living within a mutualistic consortium is that both partners adapt to each other. As a result, interacting individuals frequently evolve obligate metabolic dependencies on their corresponding counterparts, eventually even losing the capacity to survive outside the interaction[7–10]. By evolving such obligate metabolic dependencies, the evolutionary fate of interacting individuals is coupled. Even though it seems intuitively clear that living within an obligate cooperative mutualism should strongly affect the evolution of the interacting individuals, it remains generally unclear how and in which direction selective processes will be modified. Two possibilities are conceivable.

First, obligate mutualistic cooperation between two or more individuals could enhance their ability to evolve. Due to the commonly very strong mutual dependence among obligate cooperators, natural selection acts simultaneously on two different genomes. Consequently, more genetic targets are available to solve a certain evolutionary problem, thus potentially enabling mutualistic interactions to adapt faster than solitary organisms[11,12]. Moreover, in horizontally transmitted mutualisms, the formation of new combinations among interaction partners can increase the variance among mutualistic consortia, which in turn could enhance their ability to evolve[13]. An empirical example, which is often interpreted as evidence for a positive effect of a cooperative mutualism on the evolvability of the mutualistic partners, is the rich adaptive radiation of angiosperms, where the coevolution with pollinators resulted in the evolution of more than 300,000 species of flowering plants[14].

Second, obligate mutualisms could also limit the potential of genotypes to respond to environmental selection pressures. Given that the ability of cooperative interactions to survive stressful conditions likely differs between interaction partners, the individual with the lowest fitness might constrain the survival of the whole consortium (i.e. weakest link hypothesis[15]). Consequently, the niche space available to the whole association is reduced to an intersecting subset of the niche space accessible to the participating individuals[16,17]. While some studies find indeed evidence that the survival of the whole interdependent consortium can be limited by the temperature tolerance of one of its partners[18,19], others report that mutualistic interactions can ameliorate environmental stress and even extend a species' range limit[20]. Given that anthropogenic disturbances increasingly affect both terrestrial and aquatic habitats (e.g. climate change, pollution, etc.) and pose a threat to biodiversity on a global scale[21,22], a general theory of how mutualistic dependencies among individuals affects their evolutionary response to environmental stress is indispensable for predicting the broad consequences of global change for ecological communities.

However, studying how cooperative mutualisms affect the evolutionary capacity of the genotypes involved to cope with environmental stress is difficult, because the obligate nature of the interaction frequently thwarts an experimental manipulation of these systems. This is why most studies so far have addressed this question using comparative approaches[23–25] that do not allow to infer causal relationships. The ideal way to address this question would be to subject organisms, which engage in a cooperative mutualism, to an orthogonal selection pressure and compare their ability to adapt to the response of genetically identical individuals that live independently of their partner.

Here we take advantage of a laboratory-based model system consisting of two bacterial genotypes that previously evolved a costly cooperative mutualism (Fig. 1)[26]. Serial cocultivation of two auxotrophic strains, which were both unable to autonomously produce a certain amino acid and could only grow, when they reciprocally exchanged essential amino acids, favoured the evolution of cooperative genotypes that produced increased amounts of the traded metabolites. In contrast, amino acid-supplemented monocultures of auxotrophs, which were propagated in the same way, did not show this pattern. Positive fitness-feedbacks within multicellular clusters immediately rewarded an increased cooperative investment in either partner and could thus explain the evolution of mutualistic cooperation[26]. Importantly, the cooperative mutualism evolved in this study was highly beneficial when strains were allowed to grow in amino acid-deficient environments, yet incurred significant fitness costs in the absence of these benefits.

To quantify the ability of individuals that engage in an obligate cooperative mutualism to adapt evolutionarily to environmental stress and compare their responses to the ones of their free-living counterparts, both consortia of evolved cooperators and monocultures of the corresponding auxotrophs were subjected to gradually increasing concentrations of antibiotics (Fig. 1).

The results of this work show that cooperating bacteria are more susceptible to environmental stress and are less able to adapt to adverse conditions than physiologically autonomous monocultures. Moreover, we demonstrate that albeit a synergistic coevolutionary history can help cooperative strains to deal with a current evolutionary challenge, it represents a burden when the benefit of the interaction is experimentally removed. Finally, upon exposure to high antibiotic concentrations, auxotrophic bacteria engaging in an obligate mutualism display an increased tendency to regain metabolic autonomy and thus escape the obligate interaction. Together, our results identify limited evolvability as a significant evolutionary cost that individuals have to pay when entering into obligate mutualistic cooperation.

## Results

**Experimental design.** Consortia of auxotrophic *E. coli* genotypes, which previously evolved an obligate mutualistic cooperation[26], were used to determine how this type of interaction affects the ability of the participating individuals to respond to environmental selection pressures. To this end, two main experimental treatment groups were established. First, each of the two cooperative auxotrophs was grown as amino acid-supplemented monoculture (i.e. tyrosine and tryptophan, 100 μM each). Second, both genotypes were cocultivated in the absence of amino acid supplementation. A treatment, in which monocultures were cultivated in the absence of amino acid supplementation was not included, because auxotrophic genotypes would not grow under these conditions. Also, an amino acid-supplemented coculture was not implemented in the experimental design, because competition between both auxotrophs was likely to result in a loss of one of the two genotypes (Supplementary Fig. 1). Moreover, previous experiments showed that amino acid supplementation does not completely abolish the mutualistic interaction. Hence, the experiment compared monocultures with externally provided amino acids (i.e. no mutualism) to cocultures, which could only grow when strains reciprocally exchanged amino acids (i.e. mutualism). Replicate populations of both treatment groups were serially propagated while being subject to a stepwise increasing concentration of one of four different antibiotics (i.e. ampicillin, kanamycin, chloramphenicol, and tetracycline) (Fig. 1). These four antibiotics differed in their mode of action. In this way, not just the effect of a single stressor was probed, but rather the ability

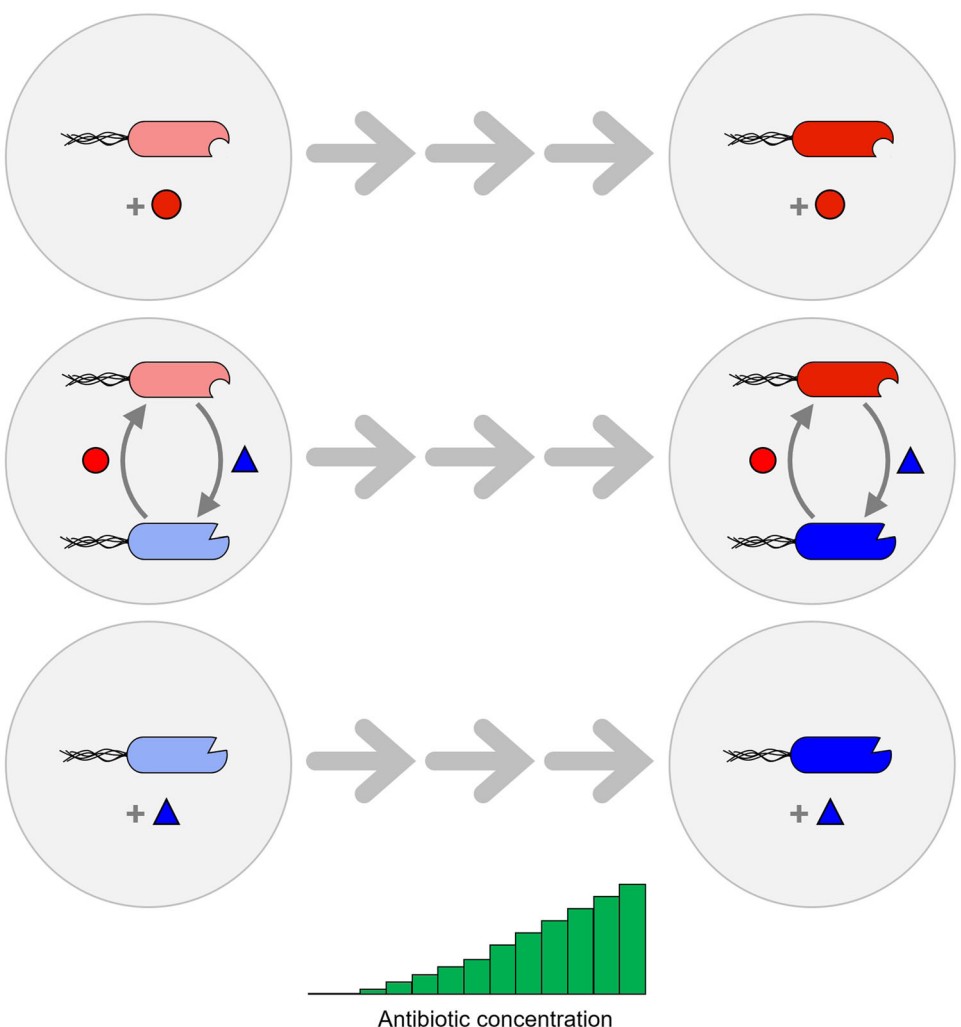

Antibiotic concentration

**Fig. 1 Design of the evolution experiment.** In a previous study, serial coevolution of two bacterial genotypes of *Escherichia coli*, which were auxotrophic for the two amino acids tryptophan ($\Delta trpB$, red cells) or tyrosine ($\Delta tyrA$, blue cells), had resulted in the evolution of obligate mutualistic cooperation between both cell types[27]. One of these mutualistic consortia was used in the current study. The two strains were either grown together in coculture (minimal medium) or in individual monocultures (minimal medium + required amino acid, 100 μM each, red circle, blue triangle). Initial populations, which were sensitive to the four different antibiotics ampicillin, kanamycin, chloramphenicol and tetracycline, were serially propagated for 15 transfers, during which the concentration of these four antibiotics was gradually increased.

of mutualistic consortia to adapt to environmental stress in general.

**Ancestral consortia differ in their growth levels and susceptibility to environmental stress.** Before the actual evolution experiment was performed, both growth levels and susceptibility to environmental stress was determined in the ancestral consortia. Comparing the maximum growth rate and densities populations achieved after 72 h revealed that unsupplemented cocultures grew significantly slower (Benjamini–Hochberg correction: $P < 0.05$, Supplementary Fig. 2a and Supplementary Table 1) and to a lower density (Benjamini-Hochberg correction: $P < 0.05$, Supplementary Fig. 2b and Supplementary Table 1) than both monocultures and cocultures that have been supplemented with amino acids (Supplementary Fig. 2).

Next, the MIC of all four antibiotics was quantified for unsupplemented monocultures as well as for both mono- and cocultures, to which amino acids have been supplied. The results of this experiment confirmed consistently for all four antibiotics tested that the MICs differed significantly between treatment groups (Supplementary Fig. 3). While unsupplemented cocultures were least

resistant to ampicillin and kanamycin of all four groups analysed, monocultures of the tryptophan-auxotrophic genotype showed the lowest resistance levels when the two antibiotics chloramphenicol and tetracycline were considered (Benjamini–Hochberg correction: $P < 0.05$, Supplementary Fig. 3c, d and Supplementary Table 2). In addition, amino acid supplementation caused a significant increase in coculture resistance levels (Benjamini–Hochberg correction: $P < 0.05$, Supplementary Fig. 3 and Supplementary Table 2) with the magnitude of this effect depending on the type of antibiotic considered. These results show that both the genotype (i.e. the identity of auxotrophy-causing mutation) and amino acid supplementation affected growth and antibiotic resistance levels of ancestral populations, thus providing a baseline for the results of the evolution experiment.

**Obligate mutualistic cooperation limits the ability of strains to adapt to environmental stress.** Analysing changes in population densities ($OD_{600nm}$) of both mono- and cocultures throughout the evolution experiment indicated that the presence of antibiotics in the growth environment had a stronger growth-reducing effect on obligate mutualistic cocultures than on

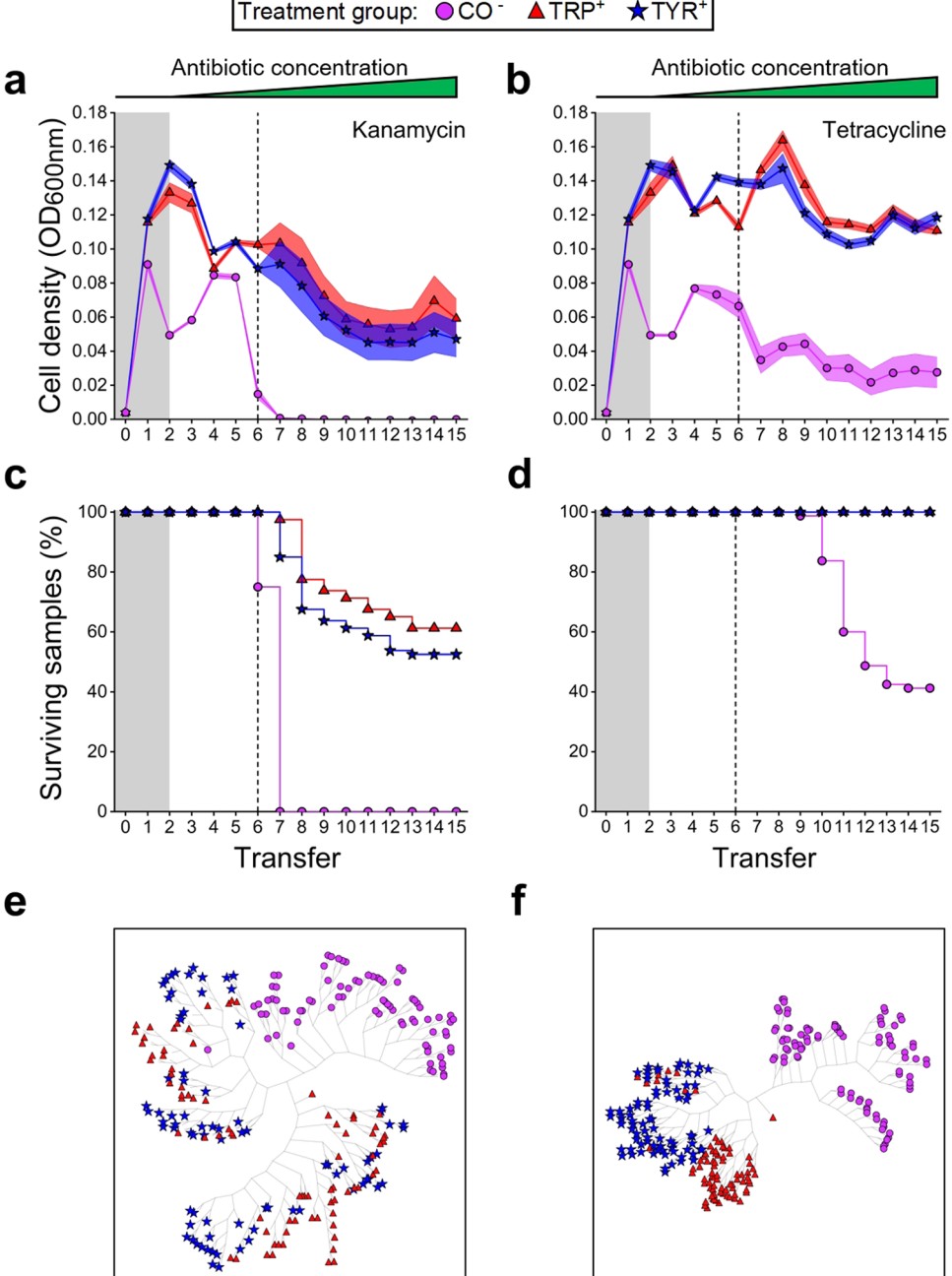

**Fig. 2 Mutualistic cooperation limits the ability of strains to adapt to environmental stress. a**, **b** Mean growth (±95% confidence interval, $n = 80$ per point) quantified as $OD_{600nm}$ and **c**, **d** proportion of surviving replicates in percent ($n = 80$ per strain) of auxotrophic monocultures (TRP, TYR) and mutualistic cocultures (CO, purple circles) of the tryptophan (TRP, red triangles) and tyrosine (TYR, blue star) auxotrophic strains throughout the evolution experiment. Antibiotic concentrations were increased in a stepwise manner after each transfer (i.e. every 72 h) (Supplementary Fig. 5). Grey-shaded areas indicate periods without antibiotic treatment. The green triangles above represent the increasing antibiotic concentrations in the evolution experiment (**a**–**d**). **a**, **c**, **e** kanamycin treatment, **b**, **d**, **f** tetracycline treatment. **e**, **f** Clustering trees of cell density profiles of experimental cultures across transfers indicate differences in the evolutionary trajectories taken by the different populations. Each leaf within a given tree represents a replicate ($n = 80$ per strain). A radial embedding layout was used to display trees. For exact $P$-values, see Supplementary Table 3. Source data are provided as a Source Data file.

monoculture controls, which were able to grow independently (Fig. 2a, b and Supplementary Fig. 4a, b). The only exception to this pattern was the monoculture of the tyrosine auxotroph that nearly went extinct upon treatment with ampicillin (Supplementary Fig. 4c).

To further analyse differences between treatment groups, the survival of cultures in the evolution experiment was compared by

applying log-rank tests for each pair of cultures. This test revealed significant differences between mutualistic consortia and monoculture controls (log-rank test: $P < 0.001$, Fig. 2c, d, Supplementary Fig. 4c, d and Supplementary Table 3), suggesting that mutualistic cocultures were more likely to go extinct than monocultures of auxotrophs. Upon reaching sub-MIC levels after the sixth transfer, a clear difference between the two bactericidal

antibiotics (ampicillin, kanamycin) and the bacteriostatic agents (chloramphenicol, tetracycline) emerged. While bactericidal antibiotics drove all cooperative cocultures to extinction (Fig. 2c and Supplementary Fig. 4c), a subset of all cultures treated with bacteriostatic agents survived until the end of the antibiotics ramping experiment (Fig. 2d, Supplementary Fig. 4d). In all four antibiotics used, cocultures showed a significantly increased death rate compared to monocultures of both tryptophan- and tyrosine-auxotrophic genotypes (log-rank test: $P < 0.001$, Fig. 2c, d, Supplementary Fig. 4c, d and Supplementary Table 3). When the two monocultures were statistically compared with each other, a significant difference was only observed for ampicillin-treated cultures, but not the groups treated with the other three antibiotics (log-rank test: $P < 0.001$, Supplementary Fig. 4c and Supplementary Table 3).

Finally, an unsupervised learning algorithm was applied to identify differences and similarities in the evolutionary trajectories of monocultures and cocultures. For this, changes in cell densities over time were used to statistically compare the different experimental groups. This analysis revealed in all cases the emergence of clusters that almost exclusively consisted of coculture replicates (Monte-Carlo resampling after $n = 10^6$ permutations: $P < 10^{-6}$, Fig. 2e, f, Supplementary Fig. 4e, f). This observation suggests cocultures followed a distinct evolutionary path that was significantly different from the trajectories of monoculture controls. For the monocultures treated with kanamycin, chloramphenicol, and tetracycline, the algorithm consistently detected clusters composed of a mixture of tryptophan and tyrosine auxotrophic monocultures (Fig. 2e, f and Supplementary Fig. 4f), while in the case of ampicillin, a clearly separated set of two clusters emerged (Monte-Carlo resampling after $n = 10^6$ permutations: $P < 10^{-6}$, Supplementary Fig. 4e). Taken together, these results clearly show that mutualistic cooperation limits the ability of obligate mutualisms to adapt to environmental selection pressures.

**Strain-level differences cause the increased susceptibility of cooperative consortia to environmental stress.** As a next step, we asked whether or not the reduced ability of cooperative consortia to cope with antibiotic-mediated selection was due to differences between individual genotypes. To address this issue, the MIC of the focal chloramphenicol- or tetracycline-treated mono- and cocultures was determined for both ancestral and derived consortia. Subtracting the MIC values of ancestral populations from the ones achieved by their derived counterparts provided a measure of how much this parameter has changed over the course of the evolution experiment. Even if the concentrations of antibiotics tested exceeded the levels strains have experienced in the original experiment, differences in the resistance of populations are indicative of their evolutionary potential: reaching a higher MIC requires mutants that are able to survive under these conditions.

The results of this analysis revealed that the increase in resistance of consortia that coevolved in unsupplemented medium was significantly lower than the one of monocultures that have evolved in the presence of amino acids (Benjamini–Hochberg correction: $P < 0.05$, Fig. 3 and Supplementary Table 4). The only exception to this was the case of the monoevolved tyrosine auxotroph, whose MIC for tetracycline increased to a significantly lower extent than was the case for the corresponding cocultures (Fig. 3b and Supplementary Table 4). Next, the MIC values from isolated individual genotypes of coevolved consortia, cultivated in the presence of the required amino acid, were compared to the ones of amino acid-supplemented cocultures. This analysis showed for both antibiotics that the MIC values of the coevolved tyrosine auxotroph increased much less than the one of the corresponding coculture,

while no such difference could be detected in the case of the tryptophan auxotrophic mutants (Fig. 3a,b). Notably, the positive effect of amino acid supplementation on antibiotic resistance levels, which was observed in ancestral cocultures (Supplementary Fig. 3), disappeared over the course of the experiment in both the chloramphenicol (Fig. 3a) and the tetracycline-treated group (Fig. 3b). This finding indicates that the metabolic intertwining between cocultured auxotrophs has tightened and that they became less able to use environmentally available amino acids.

Finally, comparing how resistance levels changed over the course of the evolution experiment between monocultures of auxotrophs that did or did not evolve as part of a mutualistic consortium revealed for all four cases analysed a significantly reduced increase of the MIC in coevolved strains relative to the corresponding monoevolved cultures (Fig. 3). The consistent response of auxotrophic strains in this experiment likely explains the reduced resistance levels coevolved consortia reached when treated with antibiotics (Fig. 3). Together, these results demonstrate that differences in the ability of individual strains to adapt to environmental stress limited the survival and thus evolvability of the entire consortium.

**Previous coadaptation to environmental stress limits the evolvability of mutualistic consortia.** Given that obligate metabolic cooperation can constrain adaptive evolution (Figs. 2 and 3 and Supplementary Fig. 4), we asked how a previous coevolutionary history in a stressful environment affects the ability of a mutualistic consortium to adapt to the same environmental challenge. We hypothesised that a shared coevolutionary history should enhance the ability of derived cocultures to cope with a previously experienced environmental stress when individuals interact with each other, yet limit their ability to tolerate increased stress levels when the otherwise obligate interaction is experimentally uncoupled. To test this hypothesis, pairwise consortia consisting of either the two coevolved auxotrophs or the two corresponding monoevolved genotypes that have been previously exposed to the two antibiotics tetracycline and chloramphenicol were again subjected to continuously increasing concentrations of the same two antibiotics. This time, however, the experiment was performed by cultivating both coevolved and monoevolved strains as cocultures (i.e. $CO_{EVO}$ and $CO_{AUX}$, Supplementary Table 5) in both the absence and presence of environmentally-supplied amino acids. This experimental design allowed us to experimentally disentangle the effect of a shared coevolutionary history (i.e. coevolved versus monoevolved) from effects emanating from the interaction itself (i.e. with versus without environmentally-supplied amino acids).

This experiment showed that in the absence of amino acid supplementation, coevolved cocultures of obligately cooperating genotypes were significantly better able to cope with the antibiotic to which they have been previously exposed (linear mixed model for chloramphenicol and tetracycline: $P < 0.001$, Fig. 4a, d and Supplementary Table 6) than cocultures of strains that previously had adapted individually to the corresponding antibiotic ($CO_{AUX}$, Supplementary Table 5). However, when the obligate dependence between bacterial partners was relaxed by externally providing the required amino acids, the observed pattern changed to the opposite. Under these conditions, individually evolved strains reached significantly higher population densities than coevolved strains (linear mixed model for chloramphenicol and tetracycline: $P < 0.001$, Fig. 4b, e and Supplementary Table 6). In other words, coevolved cooperators were better off when survival depended on a metabolic interaction between both strains, while monoevolved strains had an advantage when the need to interact was experimentally removed.

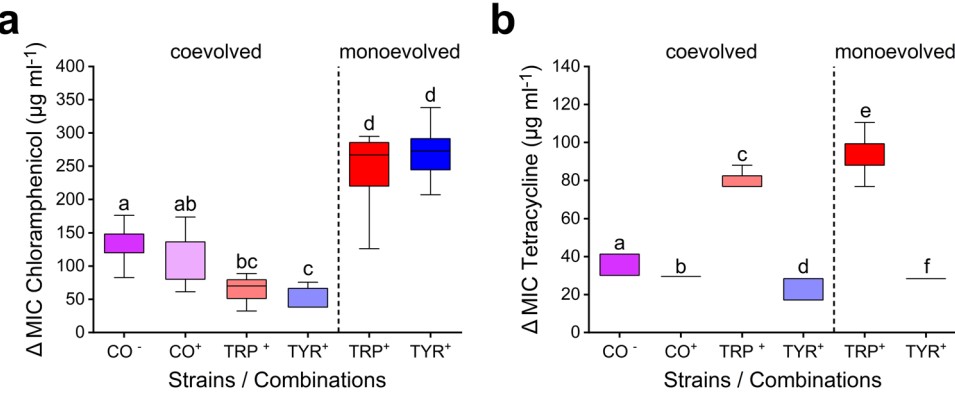

**Fig. 3 Strain-level differences cause increased susceptibility of cooperative consortia to environmental stress.** Minimum inhibitory concentration (MIC) values of ancestral (Supplementary Fig. 3) and derived strains and consortia that had evolved in the presence of **a** chloramphenicol and **b** tetracycline were assessed. ΔMIC is the difference between both values, thus indicating the increase in resistance over the course of the evolution experiment. Cocultures (CO, purple boxes) and monocultures of the tryptophan (TRP, red boxes) and tyrosine auxotrophs (TYR, blue boxes) of coevolved and monoevolved populations were analysed with ($^+$) or without ($^-$) amino acid supplementation (100 μM each). Box plots show median values (horizontal line in boxes) and the upper and lower quartiles (i.e. 25–75% of data, boxes). Whiskers indicate the 1.5x interquartile range. Different letters above boxes indicate significant differences between groups (two-sided Mann–Whitney $U$-test followed by Benjamini–Hochberg correction: $P < 0.05$, $n = 8$). For exact $P$-values, see Supplementary Table 4. Source data are provided as a Source Data file.

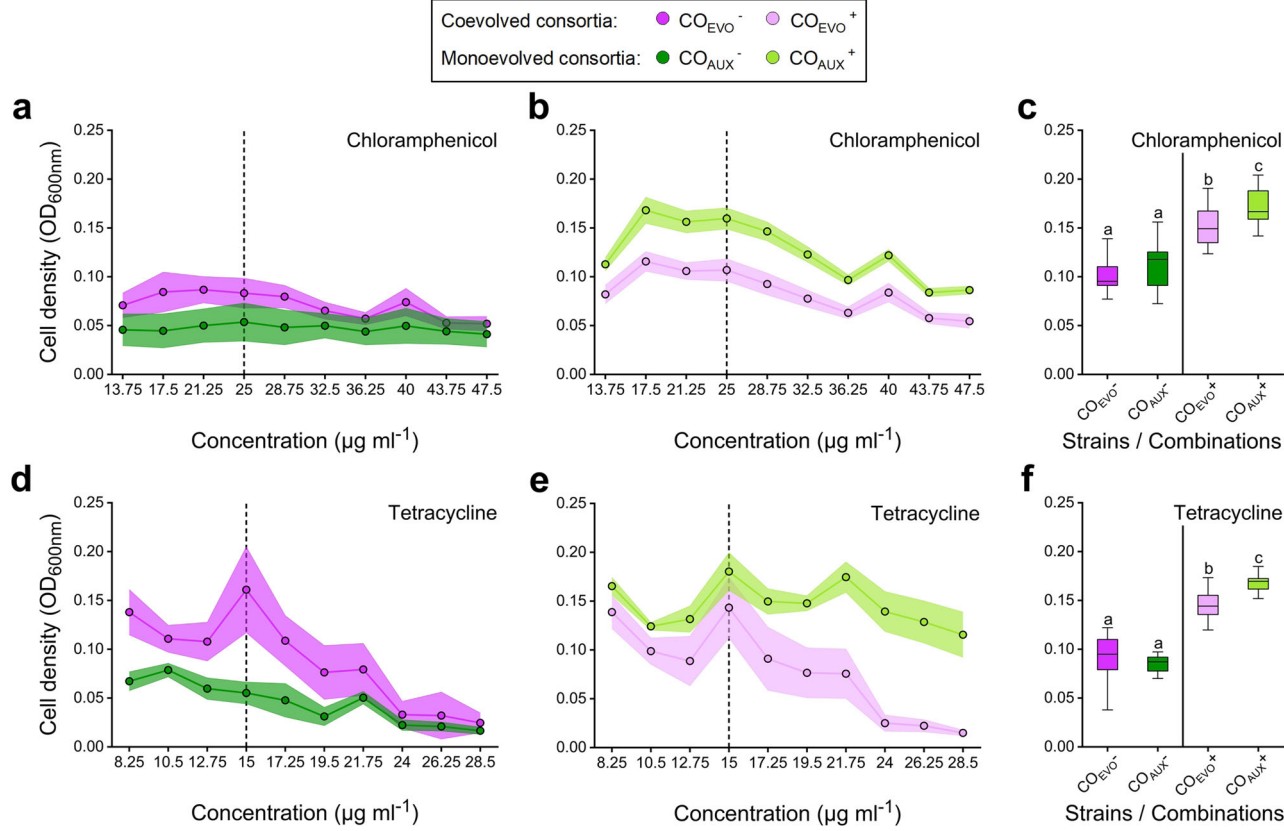

**Fig. 4 Both coevolutionary history and mutualistic dependence affect the ability of cooperative mutualists to adapt to environmental change.** Shown is the growth of coevolved auxotrophs (CO$_{EVO}$, purple circles and boxes) and cocultured monoevolved auxotrophs (CO$_{AUX}$, green circles and boxes) determined as population density (OD$_{600nm}$) with ($^+$) or without ($^-$) supplementation of tryptophan and tyrosine (100 μM each). The antibiotic, to which the respective consortia have been exposed in the evolution experiment, is indicated in each panel. Cultures were grown with **a**, **b** chloramphenicol, **d**, **e** tetracycline or **c**, **f** not treated with any antibiotic. The dashed lines mark the typical working concentration of the respective antibiotic. Data is shown as (**a**, **b**, **d**, **e**) mean (±95% confidence interval) or as **c**, **f** box plots with median values (horizontal line in boxes) and the upper and lower quartiles (i.e. 25–75% of data, boxes). Whiskers indicate the 1.5x interquartile range. Different letters above boxes indicate significant differences between groups (two-sided Mann–Whitney $U$-test followed by Benjamini–Hochberg correction: $P < 0.05$, $n = 16$). For exact $P$-values, see Supplementary Table 6. Source data are provided as a Source Data file.

Statistically comparing the growth response of consortia of coevolved genotypes when cultivated in the absence and presence of amino acids suggested that coevolved bacteria did not benefit from an external supplementation with amino acids (linear mixed model for chloramphenicol: $P = 0.067$ and for tetracycline $P = 0.74$, Fig. 4 and Supplementary Table 6). In contrast, the growth of individually evolved auxotrophs in the presence of elevated antibiotic concentrations strongly increased upon amino acid supplementation (linear mixed model for chloramphenicol and tetracycline: $P < 0.001$, Fig. 4 and Supplementary Table 6). A control experiment, in which all experimental groups were cultivated without any antibiotic treatment, showed a similar pattern, yet with less pronounced effects (Fig. 4c, f). In the absence of amino acid supplementation, both cocultures reached comparable cell densities, while growth levels of cocultures of monoevolved strains exceeded the ones of coevolved genotypes when amino acids were externally supplied (Benjamini–Hochberg correction: $P < 0.05$, Fig. 4c, f and Supplementary Table 6). Together these findings imply that coevolution curtailed the ability of strains to exist outside the interaction. This result is consistent with the interpretation that an increased dependence among genotypes coupled their evolutionary fate, thus limiting their evolvability.

**Increasing environmental stress can destabilise obligate mutualistic cooperation.** In situations where it is costly to obligately interact with another individual, for example when growth and survival depends on the amount of amino acids that the corresponding interaction partner provides, natural selection should favour individuals that evolve metabolic independence. To assess whether this also happened in the course of the antibiotics ramping experiment, the population-level proportion of mutants that evolved metabolic autonomy (i.e. reverted to a prototrophic phenotype) was quantitatively determined. For this, terminal populations of both coevolved and monoevolved auxotrophs, which have been exposed to increasing concentrations of bacteriostatic antibiotics, were randomly chosen to assess the population-level fraction of reverted phenotypes. This screening revealed that a subpopulation of the initial tyrosine auxotrophic genotypes regained the ability to grow without tyrosine supplementation. Interestingly, the rate of phenotypic reversion was significantly increased in strains that evolved in coculture relative to the respective monocultures (two-sided Pearson $\chi 2$ test for chloramphenicol TYR revertants: $P < 0.001$, $\chi^2 = 110.63$, df = 1 and for tetracycline TYR revertants: $P < 0.001$, $\chi^2 = 119.18$, df = 1, Fig. 5). In contrast, no revertants were detected among 540 screened colonies derived from 60 populations of monocultured and cocultured tryptophan auxotrophic strains. Particularly striking was the observation that no tryptophan-auxotrophic mutants could be detected in any of the tetracycline-treated cocultures (Fig. 5). This, indicates that the newly evolved phenotypic revertants either outcompeted their interaction partner or simply outnumbered them (lower detection limit: $2.5 \times 104$ cells ml$^{-1}$). Together, these results confirm that under conditions that increase the cost of mutualistic cooperation, natural selection will favour autonomous types that abandon the obligate interaction. As a result, cooperative interactions are lost from populations, thus favouring metabolically autonomous types.

## Discussion

Mutualistic interactions significantly affect the Darwinian fitness of the organisms involved[27–29]. However, it remains largely unclear how the tight ecological coupling between interaction partners affects their ability to adapt to environmental

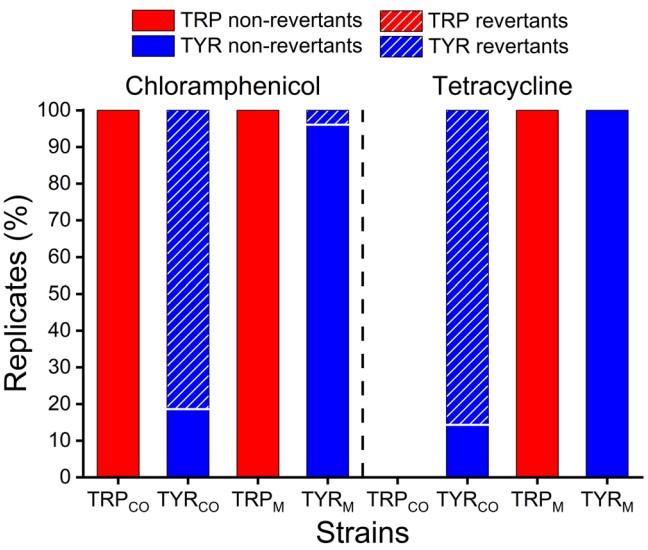

**Fig. 5 Environmental stress favours reversion to metabolic autonomy.** Shown is the population-level proportion of initially auxotrophic genotypes that evolved in the presence of chloramphenicol (left) and tetracycline (right), which remained auxotrophic (filled bar) or reverted to prototrophy (hatched bars). Tryptophan auxotrophic strains (TRP) are depicted in red and tyrosine auxotrophic strains (TYR) in blue. Populations of monoevolved genotypes (TRP$_M$, TYR$_M$) and genotypes isolated from coevolved cultures (TRP$_{CO}$, TYR$_{CO}$) are compared. The plotted colony counts (%) are the number of colonies analysed per strain relative to the total number of colonies tested in the respective cultures. In both treatments, the reversion rates of coevolved tyrosine auxotrophs were significantly higher than those of their monoevolved counterpart (chloramphenicol-treated TYR two-sided Pearson $\chi^2$ test: $P = 5.4 \times 10^{-70}$, $n = 240-285$; tetracycline-treated TYR two-sided Pearson $\chi^2$ test: $P = 4.8 \times 10^{-92}$, $n = 240-342$). Source data are provided as a Source Data file.

change[30–32]. Here we addressed this issue using an experimental evolution approach. Specifically, two bacterial genotypes that previously evolved an obligate cooperative cross-feeding interaction were exposed to a continuously increasing concentration of one of four antibiotics. The evolutionary responses of these cocultures were compared to the ones of monocultures of the same two genotypes, which were cultivated by providing them with the required amino acids. The results of these experiments revealed that (i) individuals participating in obligate mutualistic cooperation are less able to adapt to environmental stress than metabolically independent organisms (Figs. 2 and 3 and Supplementary Fig. 4), (ii) coevolution limits the ability of interaction partners to respond to changing environmental selection pressures (Fig. 4), and (iii) stressful environments can select against obligate mutualisms and favour the evolution of metabolic autonomy (Fig. 5).

The main aim of this study was to distinguish between two competing hypotheses on how being part of an obligate cooperative mutualism affects the evolutionary potential of the organisms involved. First, due to the previously reported enhanced rate of evolution[23,26,33], synergistically interacting bacteria could be better able to adapt to stressful environments. In addition, members of a mutualistic consortium might directly benefit from adaptations of their corresponding partners. For example, mutations causing resistance in one individual could also enhance the resistance levels of its interaction partner (e.g. via cross-protection). Alternatively, engaging in an obligate cooperative mutualism could constrain the ability of the participating individuals to respond to environmental change with

evolutionary adaptation[34]. The presented experimental evidence clearly supported the second hypothesis.

In our experiments, mutualistic cocultures were generally less able to adapt to environmental challenges than metabolically autonomous monocultures (Fig. 2 and Supplementary Fig. 4). One explanation for these differences could be the tight ecological coupling that evolved within consortia of mutualistically interacting bacteria. The combination of strains used here was derived from a previous study, in which two auxotrophic mutants evolved a mutualistic cooperation at a cost to themselves[26]. The resulting strains showed a reduced ability to utilise environmental amino acids and relied on a supply of these essential nutrients from their corresponding partner. This rather extreme form of physiological dependence, which likely exceeded a mere exchange of amino acids, limited the growth of the entire consortium. Thus, the resulting metabolic and physiological dependence decelerated growth and hampered the ability of the whole consortium to adapt fast enough to a rapidly increasing environmental selection pressure.

An alternative explanation for the differential ability of amino acid supplemented monocultures and unsupplemented cocultures to adapt to environmental selection pressures could be their growth levels, which differed at the onset of the experiment (Fig. 2a, b and Supplementary Fig. 4a, b). Reduced growth might limit the supply of mutations, which in turn could restrict the evolvability of mutualistic consortia. However, two main lines of reasoning suggest that this was likely not the case. First, differences in population sizes were marginal given the exponential growth of bacterial populations (i.e. ~25%). Moreover, in three out of four cases did the size of coculture populations closely approach the density reached by the corresponding monocultures of auxotrophs over the course of the evolution experiment (Fig. 2a and Supplementary Fig. 4a, b). Second, it has been shown that the mutation rate of *E. coli* is density-dependent and higher in smaller populations[35]. Accordingly, cocultures of cooperative auxotrophs should have accumulated more mutations and thus should have adapted faster than the denser populations of monocultures. Finding the opposite pattern corroborates the interpretation that the reduced ability of cooperative mutualists to respond to environmental stress by evolutionary adaptation was due to their ecological interaction and not because of growth differences between experimental groups.

Our data did not provide evidence for so-called cross-protection mutualisms, in which resistant strains protect co-occurring sensitive strains in their environment, thus allowing them to grow in the presence of antibiotics[29,36,37]. In particular, the close physical contact among interacting individuals of the focal model system could have facilitated cross-protection. However, one main difference to previous studies, which found evidence for cross-protection, is that the corresponding experiments were already initiated with antibiotic resistant genotypes[29]. In contrast, all strains used in our study were initially sensitive to all antibiotics and, thus, had to evolve resistance de novo.

In addition, our results did not support the weakest link hypothesis as proposed by Adamowicz et al. The only case, in which a behaviour matching the predictions of this hypothesis was observed in our study, was the one of ampicillin-treated cultures (Supplementary Fig. 4a). In this case, both cocultures and monocultures of the tyrosine auxotroph went almost simultaneously extinct, suggesting that the pairwise consortium might have collapsed due to the lower resistance levels of the tyrosine auxotroph. However, both the survival analysis and the analysis of the clustering tree clearly showed that the ampicillin-treated coculture and the tyrosine auxotrophic strain followed a different evolutionary trajectory (Supplementary Fig. 4c, e). Furthermore, it was found that cocultures, which coevolved in the presence of

bacteriostatic antibiotics, generally responded with a stronger increase in resistance levels to the focal compound (i.e. chloramphenicol or tetracycline) than the coevolved tyrosine auxotrophic strain when grown under monoculture conditions (Fig. 3, 'coevolved'). This finding is at odds with expectations of the 'weakest link' hypothesis, which predicts the antibiotic resistance of a given consortium should be determined by the member with the lowest resistance[15]. Together, these results clarify that the weakest link hypothesis is unlikely to be the main factor explaining the observed evolutionary patterns.

Another side effect of the harsh treatment was that these environmental conditions favoured mutants that reverted to metabolic autonomy, thus allowing them to escape the obligate relationship (Fig. 5). Under the conditions of our experiment, the cost of the interaction likely exceeded the benefits derived from it, thus selecting against obligate cooperation and favouring evolutionary independence. In fact, the evolutionary transition from obligate mutualistic cooperation to a free-living state is one possibility of how mutualisms can break down[25]. In phylogenetic analyses of obligate mutualistic interactions, free-living taxa are frequently nested within ancestrally mutualistic clades[25,38,39]. However, in many cases, the mechanistic explanation for the observed pattern remains elusive. Our study provides a possible answer: strong environmental selection pressures can challenge the ability of an obligately mutualistic consortium to adapt to changing environmental conditions, thus creating an incentive for mutants to abandon the interaction. Two possible pathways are conceivable of how a strain could have regained prototrophy in our experiments. First, it could have obtained a working copy of the previously deleted biosynthetic gene via horizontal gene transfer from its mutualistic partner. Second, it could have co-opted the biosynthetic capabilities of another gene[40]. However, future work is necessary to unravel the genetic basis of the observed compensatory evolution.

The breakdown of mutualistic interactions in response to environmental stress, as observed in our study, is likely not restricted to microbial systems, but probably also applies to other types of cooperative mutualisms. An example is the interaction between reef corals and photosynthetic zooxanthellae, in which the increasing pressure of global warming and ocean acidification frequently results in a loss of symbiotic algae from coral tissues (i.e. coral bleaching)[41,42]. Moreover, the results of our work have also important ramifications for the gut microbiota associated with humans or livestock. Studies have shown that these strongly interconnected gut microbiomes are essentially involved in a multitude of processes, including the harvesting of inaccessible nutrients and regulating the host's intestinal homeostasis and immune responses[43–45]. Treating these microbial communities with antibiotics might therefore disturb their taxonomic composition and actively select against obligate (cooperative) metabolic interactions. Given the tremendous importance of these interactions for host health[43–45], detrimental effects resulting from a loss of previously evolved metabolic interactions might impair host fitness and wellbeing for extended periods of time. Future work should assess how the treatment with antibiotics affects functional properties of gut microbiota and determine, whether potential side-effects can be neutralised, e.g. by faecal transplants[46].

Together, our results show that consortia engaging in obligate mutualistic cooperation are less able to respond to environmental selection pressures with evolutionary adaptation than physiologically autonomous individuals. Moreover, the resulting evolutionary cost of obligate cooperation can favour the emergence of genotypes capable of independent growth, thus destabilising mutualistic interactions. These results advance our understanding of the evolutionary consequences resulting from obligate cooperative interactions. Thus, the

gained insights contribute to the development of a theoretical framework that can help to quantitatively predict the effects of anthropogenic alterations of natural ecosystems, ranging from climate change to the abuse of antibiotic drugs in clinical settings.

## Methods

**Strains and plasmids.** *Escherichia coli* BW25113 $\Delta trpB$ ara- $\Delta LacZ$ (hereafter: TRP) and *Escherichia coli* BW25113 $\Delta tyrA$ ara + LacZ+ (hereafter: TYR) where used as experimental strains. Both auxotrophic strains were derived from a previous study, where they have been experimentally coevolved[26]. This selection regime led to enhanced production of the reciprocally exchanged amino acids at a cost to the producing cells. The kanamycin resistance cassette (i.e. neomycin phosphotranserase II (NPTII/Neo)) in the genome of both strains was removed using the plasmid pCP20 as described in Datsenko et al.[47].

**Culture conditions.** All experiments were performed in 96 deep-well plates (maximal volume: 2 ml, Thermo Scientific Nunc) with minimal medium for *Azospirillium brasilense* (MMAB)[48] without biotin and microelements and using glucose (5 g l$^{-1}$) instead of sodium malate as a carbon source. All media components were purchased from VWR International GmbH (Darmstadt, Germany), Carl Roth GmbH & Co. KG (Karlsruhe, Germany), or Sigma-Aldrich (Darmstadt, Germany). Cultures were incubated at 30 °C under shaken conditions (200 rpm) for 72 h. To enhance comparability and sustain sufficient growth, monocultures were supplemented with 100 µM of either tryptophan (Trp) or tyrosine (Tyr).

For plating, selective lysogeny broth (LB) agar plates and selective MMAB agar plates were used. These media were supplemented with 10 g l$^{-1}$ of L-arabinose, 1 ml l$^{-1}$ of 5% triphenyltetrazolium chloride (TTC), 1 mM of isopropyl β-D-1-thiogalactopyranoside (IPTG), and 50 µg l$^{-1}$ of 5-bromo-4-chloro-3-indolyl-β-D-galactopyranoside (x-Gal) to phenotypically distinguish differentially labelled strains in coculture.

**Growth kinetics of ancestral strains and consortia.** To determine the growth of ancestral populations, growth kinetic experiments were performed with monocultures of TRP and TYR as well as cocultures of both strains (hereafter: CO). Cocultures were grown in MMAB with and without amino acid supplementation (100 µM of each tryptophan and tyrosine) and monocultures in supplemented MMAB. Growth was followed by drop-plating 15 µl of a diluted culture on selective LB agar plates at regular intervals (i.e. after 0, 3, 6, 9, 12, 15, 18, 21, 24, 30, 42, 48, 54, 66, and 72 h) and counting the corresponding number of colony-forming units (CFUs) over the course of 72 h. Maximum growth rates were determined by averaging growth curves of all replicates of the specific treatment groups and then calculating the maximal increase of CFU numbers between each of two consecutive time points. The time point at which the slope between both points was maximal was then used to determine the maximum growth rates of individual replicates. Phenotypic markers previously introduced in the test strains allowed discriminating between cocultured strains on plates.

**Evolution experiment.** During the evolution experiment, monocultures of the tryptophan- and tyrosine-auxotrophic strains were simultaneously supplemented with both tryptophan and tyrosine (100 µM each). The media used for cocultivation did not contain any externally supplied amino acid. Each of the two monocultured strains involved in the evolution experiment started from one cryogenic stock. These strains were inoculated in 5 ml MMAB supplemented with both amino acids. After incubation for 72 h, liquid cultures were adjusted to an optical density at 600 nm (OD$_{600nm}$) of 0.1 (SpectraMax microplate reader, Molecular Devices). Precultures were then used to inoculate 16 replicates for each of the three test cultures: (1) coculture of $\Delta trpB$ ara$^-$ $\Delta lacZ$ and $\Delta tyrA$ ara$^+$ lacZ$^+$ (CO), (2) monoculture of $\Delta trpB$ ara$^-$ $\Delta lacZ$ (TRP), and (3) monoculture of $\Delta tyrA$ ara$^+$ lacZ$^+$ (TYR). In the case of both monocultures, 40 µl of preculture were used to inoculate 960 µl fresh MMAB containing both amino acids (100 µM each) and for cocultures, 20 µl of each auxotroph preculture of auxotrophs (TRP and TYR) was inoculated in 960 µl fresh MMAB without amino acids. 40 µl of the resulting cultures were transferred every 72 h into a fresh medium. During the first two growth periods, no antibiotic treatment was applied to the cultures to allow populations to equilibrate and achieve homeostasis (Supplementary Fig. 5). At the second transfer, each test culture was split up into five separate technical replicates, resulting in a total of 80 samples per treatment and strain combination. Simultaneously, the antibiotic treatment started at this step. Growth of all cultures was tracked by quantifying their population density (OD$_{600nm}$), while propagating them to fresh medium. In addition, the death rate of all cultures was determined during the transfer: a culture was considered dead when the OD$_{600nm}$ value fell below the critical threshold of 0.01.

Over the course of the experiment, antibiotic concentrations were gradually increased (Fig. 1 and Supplementary Fig. 5). Two main features characterised this ramping design. First, the antibiotic concentration doubled during each transfer until it reached the determined sub-minimal inhibitory concentration (sub-MIC, Supplementary Fig. 6), starting from the second transfer with one-eighth of the respective sub-MIC. Second, after reaching the sub-MIC level, the antibiotic

increment followed a linear function until it was slightly above the respective working concentrations. The rates, at which antibiotic concentrations were increased every transfer, were individually adjusted for each antibiotic, considering their sub-MIC values and targeted working concentrations. The corresponding working concentrations were 100 µg ml$^{-1}$ for ampicillin, 25 µg ml$^{-1}$ for chloramphenicol, 50 µg ml$^{-1}$ for kanamycin, and 15 µg ml$^{-1}$ for tetracycline. This information was provided by the manufacturer (Carl Roth GmbH & Co. KG, Karlsruhe, Germany) or derived from the Addgene archive[49]. These four antibiotics were chosen to maximise differences in terms of their effect (i.e. bacteriostatic or bactericidal), mode of action, and most frequently observed resistance mechanism[50–52].

**Minimal inhibitory concentration.** Two sets of experiments were performed to determine the MIC of monocultures and cocultures. First, to adjust the antibiotic pressure of the four focal antibiotics during the evolution experiment, sub-MIC values of ancestral populations (TRP and TYR) and consortia (CO) were established for each compound before the evolution experiments has been initiated[53,54]. Second, to identify the concentration of chloramphenicol or tetracycline populations that could survive after the evolution experiment, the MIC of individual genotypes and cocultures that had adapted to both antibiotics were analysed. In this context, the two bacteriostatic antibiotics were considered, because only here a subset of replicate populations survived until the end of the main experiment (Fig. 2d and Supplementary Fig. 4d). In both cases (i.e. before and after the evolution experiment), MIC test experiments of cocultures were performed with and without amino acid supplementation (100 µM each), while monocultures were only assayed under supplemented conditions.

The sub-MIC range was defined as 0.05 ± 0.005 OD$_{600nm}$. Within this range, the selection pressure due to the antibiotic treatment is maximal, yet the growth of bacterial cultures is still sufficient to allow for sustainable growth when populations are transferred to start the next growth cycle. Hereafter, this range is referred to as the threshold zone. An antibiotic concentration was considered sub-MIC if all measured OD$_{600nm}$ values of one antibiotic concentration fell within the threshold zone (Supplementary Fig. 6). In cases, where OD$_{600nm}$ values of more than one antibiotic concentration lay in the threshold zone, their average value was considered as sub-MIC. Moreover, if only a partial overlap of the values at one antibiotic concentration and the threshold zone was observed and the next measured value was not within the range, a mean value between those two was selected as sub-MIC. Only when there was no next measurement, the current concentration was picked.

Based on these parameters, the sub-MIC values for the antibiotics used were 0.75 µg ml$^{-1}$ for ampicillin, 3.25 µg ml$^{-1}$ for chloramphenicol, 1.25 µg ml$^{-1}$ for kanamycin, and 1.35 µg ml$^{-1}$ for tetracycline (Supplementary Fig. 6). All MIC measurements were performed according to protocols of Wiegand et al.[54] and Andrews[53]. Growth depending on the current concentration of antibiotics was determined by quantifying population densities as OD$_{600nm}$. The first antibiotic concentration, at which the measured OD$_{600nm}$ value of culture did not cross a threshold of 0.01, was considered as MIC.

**Separation of coevolved cocultures.** Eight replicates from the chloramphenicol- and tetracycline-treated evolved cocultures were selected to separate individual auxotrophic genotypes from each other. For this, cultures were grown for 72 h and spread-plated on selective MMAB agar supplemented with tryptophan and tyrosine (100 µM each). Separation was performed by picking single colonies from these plates and streaking them on five different selective plates (MMAB + Trp+Tyr, MMAB, MMAB + Trp, MMAB + Tyr, LB) to determine their phenotype. From chloramphenicol-treated cocultures, three colonies belonging to each genotype (TRP and TYR) were isolated per replicate. However, for tetracycline-treated cocultures, only three colonies could be isolated in total that showed the sought tryptophan background. This was most likely due to the low abundance of tryptophan-auxotrophic genotypes in these cultures (Fig. 5).

**Effect of coevolutionary history and metabolic dependence.** An experiment was conducted to determine the interactive effect between (i) the previous coevolutionary history in a certain stressful environment as well as (ii) the metabolic dependence on another genotype and the ability of strains to cope with environmental stress. For this, strains that evolved in the presence of chloramphenicol and tetracycline either alone (i.e., monoevolved) or in coculture were grown as coculture (both consisting of either monoevolved (Co$_{AUX}$) or coevolved strains (CO$_{EVO}$), Supplementary Table 5) in the absence and presence of the two required amino acids (100 µM each). In total, eight randomly selected pairs of mono- or coevolved tryptophan and tyrosine auxotrophs were used, which have been isolated from the 15$^{th}$ transfer of the main experiment. All of these populations were cultivated at different concentrations of the antibiotics chloramphenicol and tetracycline or in the absence of antibiotic treatment and the population densities they achieved 72 h post-inoculation were quantified as OD$_{600nm}$.

**Phenotypic reversion.** To quantify the rate of phenotypic reversion within cultures that evolved in presence of chloramphenicol or tetracycline, 15 replicates from the 15$^{th}$ transfer of each test culture type (CO, TRP and TYR) were randomly

selected to determine whether the population was still auxotrophic or if also prototrophic revertants have arisen to detectable levels. Precultures of evolved monocultures were supplemented with both tryptophan and tyrosine (100 µM each), while precultures of coevolved cocultures were grown in the absence of amino acids for 72 h in liquid MMAB. In addition, all cultures were supplemented with either chloramphenicol (13.75 µg ml$^{-1}$) or tetracycline (8.25 µg ml$^{-1}$) to mimic the conditions of the evolution experiment as closely as possible, while maintaining sufficient growth. These precultures were spread-plated on selective LB agar and single colonies were picked from each test sample and streaked on four different selective plates (MMAB, MMAB + Trp, MMAB + Tyr, LB). Colonies that grew on all plates comparably well were considered as phenotypic revertants, based on the apparent loss of their auxotrophic phenotype. In this way, it was possible to differentiate and classify auxotrophic and prototrophic colonies within cocultures with a lower detection limit of $2.5 \times 10^4$ CFU ml$^{-1}$ and an upper detection limit of $2.5 \times 10^6$ CFU ml$^{-1}$.

**Statistical analysis**. An unsupervised learning algorithm was used to characterise the evolutionary trajectory followed by mono- and cocultures of auxotrophs during the evolution experiment. In particular, profiles of cell densities between cultures and across transfers were compared using a variational Gaussian mixture algorithm to detect clusters in the data, using the FindClusters routine implemented in Mathematica 12.0.0.0. To compare survival distributions between cultures of this experiment, log-rank tests were calculated for each pair of cultures using the SurvivalModelFit and LogRankTest routines implemented in Mathematica 12.0.0.0.

Differences in the evolvability of coevolved cocultures and individually evolved strains were assessed using linear mixed models with bacterial growth as the response variable. To compare strain types in the presence and absence of amino acids, contrasts between their growth levels were calculated using the *emmeans* package function *emmeans* from R[55] (Supplementary Information 1).

Differences in the level of antibiotic resistance, maximum growth rate, and yield were identified applying a two-sided Mann–Whitney *U*-test followed by the Benjamini–Hochberg correction using R[55]. Significant differences in the population-level ratio in the number of phenotypic reversion of evolved monocultures and cocultures were determined using a two-sided Pearson $\chi^2$ test in SPSS (Version 25, IBM®). Normal distribution of datasets was assessed using the Kolmogorov–Smirnov test and homogeneity of variances was verified by applying Levene's test. Variances were considered to be homogeneous when $P > 0.05$.

**Reporting summary**. Further information on research design is available in the Nature Research Reporting Summary linked to this article.

## Data availability
All data generated in this study are available as a Source Data file and in a Zenodo database (accession number: https://zenodo.org/record/5720811#.YZzmuS-B1qs). The generated strains are available from the corresponding author upon request. Source data are provided with this paper.

## Code availability
The source code used for the data analysis in this study is available in a Zenodo database (accession number: https://zenodo.org/record/5720811#.YZzmuS-B1qs).

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

## Acknowledgements

The authors thank Daniel Preußger for providing strains and advice on experiments. This manuscript benefitted greatly from discussions with Piyali Pal Chowdhury, Samir Giri, and all other members of the Kostlab. This work was funded by the German Research Foundation (SFB 944, P19: C.K. and B.P.; SPP1617, KO 3909/2-1: C.K.), the International Graduate School *EvoCell* (C.K.), and Osnabrück University (L.O.).

## Author contributions

Conceived the study: C.K. and B.P. Designed the experiments: B.P., C.K., and L.O. Performed the experiments: B.P. and M.H. Analysed the results: L.O. and B.P. Interpreted the results: B.P., L.O., and C.K. Wrote the manuscript: B.P. and C.K. Amended the manuscript: B.P., C.K., and L.O.

## Funding

## Competing interests

The authors declare no competing interests.

## Additional information

**Peer review information** *Nature Communications* thanks Stuart West and the other anonymous reviewer(s) for their contribution to the peer review this work. Peer reviewer reports are available.

