## [Peer Review File · Nature Communications]

Reviewers' Comments:

Reviewer #1:

Remarks to the Author:

The goal of this study was to investigate how metabolic interactions between two bacterial strains influence their ability to evolutionarily adapt to environmental changes- in this case, exposure to different antibiotics. Using four different types of antibiotic, the authors compare the resistance evolution patterns of monoculture-grown versus coculture-grown cultures of mutualistic auxotrophic *E. coli* strains. In agreement with previous studies, the authors find that cocultures evolve less resistance than do their monoculture counterparts. Excitingly, the authors also find that, upon re-exposure to the same antibiotic, coevolved cooperators have more robust growth than monoevolved strains grown in co-culture. However, this difference is reversed when the obligate dependence between strains is removed by adding required amino acids to the growth medium. Additionally, the authors find that phenotypic reversion of amino acid auxotrophies under antibiotic selection conditions that favor the growth of metabolically independent strains.

The work outlined in this manuscript is novel and adds greatly to a growing body of literature on the impact of cooperative interactions on bacterial evolvability. The statistical analyses are appropriate and valid, and the level of detail in the methods section is sufficient for others to reproduce the study's findings. This work will be of particular interest to microbial ecologists studying how cooperation influences evolution, and to microbiologists interested in factors impacting antibiotic resistance evolution. Overall, this paper is excellent and makes an important contribution to several fields.

Major comments:

- In figure 2A and B, and in supplemental figure 1A and B, the co-cultures show a drop in OD600 at the second passage, followed by a period of increasing OD600 prior to the antibiotic concentration exceeding MIC values. This increase in OD600 at sub-lethal antibiotic concentrations is not seen in monocultures. Does this suggest that sub-lethal antibiotic exposure has some benefit to co-cultures?
- My understanding is that phenotypic reversion of amino acid auxotrophy is highly unusual, particularly when the auxotrophic strains are derived from knockout mutations in amino acid biosynthetic pathways, as seems to be the case here. Do the authors have any speculation as to how this occurred? Might an auxotrophic strain have horizontally acquired a functional copy of the knocked-out gene from its metabolic partner at some point during the evolution experiment? A few sentences in the discussion about the authors' speculations on this phenomenon should be added.

Minor comments:

- What are the relative growth rates of each *E. coli* strain in monoculture vs. in coculture? This may influence the experimental results, as slower growth rates can increase phenotypic tolerance to antibiotics.
- Line 52-53: It would be helpful to note here that the angiosperm partner is pollinators; non-ecologists may not be familiar with this example.
- Line 186-187: in this experiment, were the coevolved and monoevolved strains partnered with ancestral strains when grown as co-cultures? Or was, for example, a monoevolved strain paired with another monoevolved strain?

Reviewer #2:

Remarks to the Author:

The paper addresses an important question – how mutualisms influence long-term radiations, evolvability etc. Most work in this area is comparative, and this paper brings a fresh experimental approach. The introduction setup was very clear and outlined the novelty of this work. The results are interesting and show a potential long term cost of being in a mutualism.

However, I worry that the basic experimental design was unbalanced and flawed. Examining the first two results sections, there appears not to be a treatment where the two strains were co-cultured, and the required amino acid was supplied. This is my reading based on figure 1 and the

methods. This design is unbalanced because it has "+amino acid" confounded with "number of strains". It is analogous to missing a key control treatment (the 2 strains cocultured, with amino acid supplied). If I have understood correctly, this would invalidate the conclusions.

For example, the first results section is titled "Obligate mutualistic cooperation limits the ability of strains to adapt to environmental stress". But we can't separate if the result has arisen because of obligate mutualism (i.e. whether amino acids were supplied), or because of the number of strains interacting / competing (i.e. 1 or 2 / was it a coculture or a monoculture).

The third section has a new experiment, which then compares coevolved versus monoevolved. It does this with a balanced design, looking at evolution with and without amino acids. However, again the same problem arises, because the strains aren't just coevolved versus monoevolved. Their history from the first experiment is coevolved and without amino acid, versus monoevolved and with amino acid (i.e. coevolved/monoevolved was confounded with amino acid supplementation).

I feel bad pointing this out. The results make sense, and I suspect are probably 'true'. But without the appropriate control this can't be said scientifically. Control treatments are fundamental to experimental evolution – I have focused on the most basic possible, but confidence could probably have been increased even further, by trying to develop others as well.

Minor comments:

1. Introduction, first+ third paragraph. Does this also relate to Bob May's classic theory (his 73 book *Stability & Complexity in Model Ecosystems*) on how mutualism population dynamics are less stable?
2. Line 107. I know it cuts straight from intro to results, but it would be useful to provide a bit more summary of what the actual selection experiment was, to provide context to the results (to go with fig 1 – which was very clear).

Reviewer #3:

Remarks to the Author:

This is a good manuscript with valuable results. It requires few modifications, as some points need further explanations, and some aspects could be explained more briefly. Some weak points of the paper consist of fact that the two strains used in the experiment are treated as black boxes, with no indications on the genomics or on the metabolism of the bacteria. We do not know why the strains react in the way they do.

The authors perform experiments using two modified strains of *E. coli* to demonstrate a number of points regarding cooperation and evolvability under environmental stress.

1. Metabolically interdependent strains grow less under antibiotic stress than strains on their own.
2. Coevolution can enable or restrain the ability of interaction partners to respond to changing environmental selection pressures.
3. Under stress, mutualistic interactions can collapse and strains can regain metabolic autonomy.

Below I included observations regarding each of these points:

1. For the first point, the authors find "surprising" that "cocultures of mutualistically interacting *E. coli* strains reached lower population densities and reduced resistance levels as compared to the corresponding monoculture controls (Figs. 2, 3, and Supplementary Fig. 1)" (Discussion, Line 265). To support this argument, the authors cite studies that demonstrate mutualistic relationships of two "unequal" species, in which resistant strains can protect the sensitive ones. In the particular setup of this study, both participating strains are more or less equal to each other (both are resistant or susceptible to the antibiotics, there is no one strain more resistant than the other, and their growth rates in absence of antibiotics is not shown, but are presumed similar to each other).

A simple mathematical demonstration can show that the result of this experiment is predictable.

Let A be the fitness of the first strain and B the fitness of the second strain.
Then $A * B$ = the initial fitness of the strains A and B growing in coculture. The initial total fitness of strains A and B growing separately is $A+B$.
When exposed to antibiotics, the fitness of the two strains incurs penalties A_1 and B_1 ($0 < A_1 < 1$ and $0 < B_1 < 1$), as follows:
Fitness of A becomes $(A - A * A_1)$ and the fitness of B becomes $(B - B * B_1)$.
Thus, the new fitness of the cocultures is:
 $(A - A * A_1) + (B - B * B_1) = A * B - A * B * B_1 - A * B * A_1 + A * B * A_1 * B_1 = A * B - A * A_1 * (B - \frac{1}{2} * B * B_1) - B * B_1 * (A - \frac{1}{2} * A * A_1)$, where $A * B$ is the initial fitness, before the antibiotics are applied. (1)
The new total fitness of the strains A and B growing separately is: $A + B - A * A_1 - B * B_1$, where $A+B$ is the initial fitness. (2)
The terms in the second and third positions in statement, showing decrease of initial fitness (1) are greater than the terms showing decrease of fitness in statement (2), as $A * A_1 * (B - \frac{1}{2} * B * B_1) > A * A_1$ and $B * B_1 * (A - \frac{1}{2} * A * A_1) > B * B_1$. Thus, the strains growing in coculture are expected to have lower fitness and reach lower population densities than the strains growing in isolation, when antibiotics are administered.

2. This is a very neat experiment, with nice results. However, some details on genomic mechanisms that lead to AMR, in monocultures and in cocultures, should be included.

3. This is also a very nice result. However, the two *E. coli* strains are described like black boxes. Why and how does the Tyrosine-auxotrophic mutant regain metabolic autonomy, but the tryptophane-auxotrophic does not? An explanation regarding the strains different metabolism or genetics should be included.

Few more points:

1. Line 138 mentions an unsupervised learning algorithm to identify the difference and similarities in the evolutionary trajectories of monocultures and cocultures. This is not clear. What clusters does this refer to? It is not clear what this analysis is trying to show.
2. Line 148: The authors show that cocultures of strains that adapted together grow better than cocultures of strains that evolved separately, when exposed to TET and Chloramphenicol. This trend changes, when the coculture is supplemented with required AA: the supplemented cocultures of strains that evolved separately grow better than supplemented cocultures adapted together. Were the types of cocultures (with and without AA, adapted and not adapted) compared, in the absence of antibiotics? It appears obvious that supplementing the non-adapted coculture with AA leads to better growth than the supplemented adapted one.
3. Regarding Figure 2: It is surprising that resistance was acquired so fast. We see little drop in population density after the sub-MIC dash line in the Tetracycline plot. Why is this?
4. Regarding Figure 2: I suggest that instead of Fig2 E and F, confidence intervals could be added to Fig2 C and D.
5. Figure 3: What do the colours of the box plot mean? There is no legend. It is not clear what the letters above the box plot mean (differences between the strains).
6. In Fig 3A: Does this mean that supplementation of TRP leads to worse results than no supplementation at all (CO- versus TRP+)? Can this be clarified in text?

Reviewer #4:

Remarks to the Author:

Kost review

The author's present a clear and accessible set of experiments examining the stability and adaptability of a two-strain community stabilized by a mutualism relative to populations founded by corresponding individual strains. I think the central results are clear and strong. I do, however,

have one overarching concern – the key results are not accompanied by follow-up sufficient to determine even their general basis. This limits my ability to evaluate how they can be generalized.

Major comments:

1. I very rarely critique a manuscript for what is absent (barring specific controls). Here, though, I can't help but think that a fuller explanation of the basis of at least the main finding – that cocultures are less well able to adapt than component strains – would greatly increase the impact of the work.

By way of example – a possible mechanism for reduced coculture evolvability is that coculture populations simply reach lower densities, and thus have less mutational input, than do monocultures. Population densities recorded at the end of the first and second transfer (Fig. 2AB and S1AB) – when no antibiotic was present – show a lower density for cocultures, which would be consistent with this possibility. (They also show a big decline in coculture population size at the end of the first transfer relative to the second, suggesting that they were not well adapted to the base medium.) To isolate this effect, it would be interesting to have compared evolvability with population size controlled for (perhaps by reducing the input of supplemented amino acid to monocultures) so that the influence of strain interactions could be isolated. I wonder also if the apparently lower amount of limiting amino acid in cocultures relative to monocultures influenced the results reported in Fig. 5. If coculture strains were under stronger selection for prototrophy, because environmental amino acids were not sufficient for fast growth of auxotrophs, then a higher frequency of reversion to prototrophy would be expected. Perhaps it can be argued that this kind of effect is the point of the experiment – cocultures are costly – but the fact that the amount of amino acid added to cocultures, and thus, population sizes and strength of selection for prototrophy in monocultures, is presented here as being arbitrary, for me undermines the significance, or at least the generalizability of results.

In summary, I'd really like the authors to make some effort to speak to the basis of the main results they obtain.

2. The MIC follow-up (Fig. 3) to the evolution experiment outcome presented in Fig. 2 is interpreted to show that differences in monoevolved and coevolved population MICs likely impacted adaptability. I'm trying to reconcile the differences in scale. If I understand correctly, the evolution experiment ended at antibiotic concentrations below the ones obtained by every group plotted in Fig 3. How should we interpret, for example, that 'resistance levels reached by coevolved consortia was significantly lower than one of the monoevolved strains' when that low resistance was nevertheless above the level required for growth in the original evolution experiment. Why should MIC above the selected value be relevant to population success? A clearer assessment of this experiment to the outcomes of the main evolution experiment would be very helpful.

Minor comments:

Small typographical issues, most I haven't detailed.

L50. Explain the mutualism relevant to angiosperms as an example of a positive consequence of a mutualism for evolvability.

L62. 'extent' to 'extend'

L70. 'system' to 'systems'

L188. "allowed *us* to"

L191, Fig 4. I suggest that this analysis should account for different initial levels of adaptation to the antibiotics. My guess is that changes are conservative to this factor (e.g., in Tc monocultures start being slightly better adapted than strains isolated from cocultures, though the coculture strains go on to reach higher ODs across all tested Tc concentrations), but it would be good to be reassured.

L216. Needs more context.

L258. It would be useful to clarify how this (ecological?) effect influences evolutionary potential.

L270. I would have thought that cross-protection would depend on the nature of the resistance mechanism? E.g., Tc resistance might involve upregulation of efflux pumps, potentially creating a

relatively higher concentration of the drug around a resistant cell. Although the actual resistance mechanisms that evolved might not be known, it is probably unlikely that they would involve modification/deactivation of the drug, which is the mechanism most likely to confer cross-resistance. Perhaps this is the point the authors are getting at when they note that resistance had to evolve de novo in this work? If so, it would be useful to clarify.

Fig. 2. I think it is up to the authors, but I would suggest clarifying that the X-axis here indicates increasing antibiotic concentrations. That information is clearly in the text and legend, but interpretability of the top four panels could be helped with inset of an 'Ab concentration 'triangle' or similar.

Fig. 2 EF. To me this analysis doesn't add much – it is clear from panels A-D that cocultures are less adaptable (including associated statistical analysis). My suggestion is to move to the supplementary material.

Response to reviewer's comments

Reviewer #1

The goal of this study was to investigate how metabolic interactions between two bacterial strains influence their ability to evolutionarily adapt to environmental changes- in this case, exposure to different antibiotics. Using four different types of antibiotic, the authors compare the resistance evolution patterns of monoculture-grown versus coculture-grown cultures of mutualistic auxotrophic *E. coli* strains. In agreement with previous studies, the authors find that cocultures evolve less resistance than do their monoculture counterparts. Excitingly, the authors also find that, upon re-exposure to the same antibiotic, coevolved cooperators have more robust growth than monoevolved strains grown in co-culture. However, this difference is reversed when the obligate dependence between strains is removed by adding required amino acids to the growth medium. Additionally, the authors find phenotypic reversion of amino acid auxotrophies under antibiotic selection conditions, which favors the growth of metabolically independent strains.

The work outlined in this manuscript is novel and adds greatly to a growing body of literature on the impact of cooperative interactions on bacterial evolvability. The statistical analyses are appropriate and valid, and the level of detail in the methods section is sufficient for others to reproduce the study's findings. This work will be of particular interest to microbial ecologists studying how cooperation influences evolution, and to microbiologists interested in factors impacting antibiotic resistance evolution. Overall, this paper is excellent and makes an important contribution to several fields.

We thank the reviewer for the positive feedback.

Major comments:

(1) In figure 2A and B, and in supplemental figure 1A and B, the co-cultures show a drop in OD600 at the second passage, followed by a period of increasing OD600 prior to the antibiotic concentration exceeding MIC values. This increase in OD600 at sub-lethal antibiotic concentrations is not seen in monocultures. Does this suggest that sub-lethal antibiotic exposure has some benefit to co-cultures?

In this early phase of the evolution experiment, multiple processes operate simultaneously. First, strains adjust physiologically to the growth environment. Second, in coculture, relative strain frequencies are adjusted by negative frequency-dependent selection. Third, strains start to adapt to sub-lethal concentrations of the antibiotics. Given this complexity of interacting factors, it is difficult to pinpoint a single explanation. The observed pattern was most likely due to a combination of the abovementioned processes.

(2) My understanding is that phenotypic reversion of amino acid auxotrophy is highly unusual, particularly when the auxotrophic strains are derived from knockout mutations in amino acid biosynthetic pathways, as seems to be the case here. Do the authors have any speculation as to how this occurred? Might an auxotrophic strain have horizontally acquired a functional copy of the knocked-out gene from its metabolic partner at some point during the evolution experiment? A few sentences in the discussion about the authors' speculations on this phenomenon should be added.

Phenotypic reversion of amino acid auxotrophies have been previously reported in the literature (see e.g. Kiss and Stephanopoulos 1992 *Biotechnology and bioengineering*, Wright and Minnick 1997 *Microbiology*). Under coculture conditions, the benefit of cross-feeding likely exceeds the potential advantage of metabolic autonomy, thus favouring auxotrophic mutants over reverted genotypes. However, the cost-to-benefit ratio is likely shifted when cross-feeding auxotrophs need to adapt to an additional stressor, in this case the steadily increasing concentration of antibiotics. In the revised version of the manuscript, we included a section in the discussion in which we summarize possible genetic mechanisms that could explain the observed phenotypic reversion to prototrophy (lines 383-391).

Minor comments:

(3) What are the relative growth rates of each *E. coli* strain in monoculture vs. in coculture? This may influence the experimental results, as slower growth rates can increase phenotypic tolerance to antibiotics.

To address the point raised by the reviewer, we have performed additional experiments to determine the growth rate and MIC of ancestral strains and consortia. The resulting data has been included in the revised version of the manuscript (Supplemental Figs. 2, 3). Analysing this data did not reveal a correlation between the decreased growth rate of cocultures and their initial resistance. This clearly suggests, that the reduced ability of cross-feeding cultures to adapt to steadily increasing concentrations of antibiotics was not due to initial growth differences of the different groups (lines 127-150; Supplemental Figs. 2, 3).

(4) Line 52-53: It would be helpful to note here that the angiosperm partner is pollinators; non-ecologists may not be familiar with this example.

To clarify this point, we now mention the word “pollinators” in the corresponding paragraph (line 52).

(5) Line 186-187: in this experiment, were the coevolved and monoevolved strains partnered with ancestral strains when grown as co-cultures? Or was, for example, a monoevolved strain paired with another monoevolved strain?

We thank the reviewer for pointing out this unclarity. In this experiment, ancestral strains were not paired with evolved ones, but cocultures were assembled from two isolated strains that either coevolved previously (i.e. with coevolutionary history) or evolved as monoculture (i.e. without coevolutionary history). In the revised version of the manuscript, we have reworded this passage to clarify this point (lines 234-243). In addition, Supplementary Table 5 explains in detail, which combinations of strains have been analysed.

Reviewer #2

The paper addresses an important question – how mutualisms influence long-term radiations, evolvability etc. Most work in this area is comparative, and this paper brings a fresh experimental approach. The introduction setup was very clear and outlined the novelty of this work. The results are interesting and show a potential long term cost of being in a mutualism.

We thank the reviewer for the positive feedback to our manuscript.

However, I worry that the basic experimental design was unbalanced and flawed. Examining the first two results sections, there appears not to be a treatment where the two strains were co-cultured, and the required amino acid was supplied. This is my reading based on figure 1 and the methods. This design is unbalanced because it has “+amino acid” confounded with “number of strains”. It is analogous to missing a key control treatment (the 2 strains cocultured, with amino acid supplied). If I have understood correctly, this would invalidate the conclusions. For example, the first results section is titled “Obligate mutualistic cooperation limits the ability of strains to adapt to environmental stress”. But we can’t separate if the result has arisen because of obligate mutualism (i.e. whether amino acids were supplied), or because of the number of strains interacting / competing (i.e. 1 or 2 / was it a coculture or a monoculture).

We thank the reviewer for this important comment.

The evolution experiment analyses the factor “mutualism” and thus requires two experimental groups, in which the same bacterial genotypes do or do not engage in a mutualistic interaction. Due to the specificities of the focal study system used, it was not possible to just cultivate auxotrophic genotypes in monoculture as the “no mutualism” treatment, but an additional supplementation with amino acids was required to allow them to grow. Now, we have included a second factor “amino acid” and ideally, one would use a full factorial design, in which both experimental treatments (coculture/ monoculture and plus/ minus amino acid supplementation) are combined in all possible combinations. However, we deliberately opted against this possibility and instead chose a different experimental design. In the following, we would like to explain the reasons for this:

(i) It was biologically not possible to implement the full factorial design, because auxotrophs in monoculture that are not provided with amino acids, cannot grow at all and, thus, would readily die out. Hence, it did not make sense to include this treatment group.

(ii) The rationale for including a treatment “coculture with amino acid supplementation”, as requested by the reviewer, would have been based on the expectation that strains do not form a mutualistic interaction, because amino acid supplementation uncouples the obligate metabolic requirement. Instead, this treatment would control for the effect of amino acids on the strain’s ability to respond to the applied antibiotics treatment. However, this premise is violated. We know from many other experiments with these strains that cultivating our mutualistic consortia in the presence of amino acids does not completely abolish the mutualistic interaction. Instead, cells still form multicellular clusters, in which they reciprocally exchange essential amino acids. This happens in particular at later stages of the growth phase, when the environmentally available amino acids are depleted. Thus, including this treatment group would not have had the desired effect (i.e. remove the mutualism and control for the effects of the amino acid treatment only). Instead, it would have been unclear whether and to which extent the response of this treatment group was due to a mutualistic interaction or not. Since we knew that this treatment would not have entirely removed the mutualistic interaction before we started the actual experiment, we did not include it, because the results of this treatment group would have been very difficult to interpret.

Another reason for not including the “coculture with amino acid supplementation” was the prior observation that despite the effect described in (ii), strains also competed with each other for the supplied amino acid (Supplemental Fig. 1B). As a consequence of this, strain frequencies drastically changed, thus impairing the comparability of the data resulting from this treatment to the treatment “coculture without amino acid supplementation”, in which the ratio of strains remained constant throughout the entire period of the experiment (Supplemental Fig. 1A). Moreover, in the presence of antibiotics, the weaker competitor in the “coculture with amino acid supplementation” treatment would likely have died out due to its lower population density, in which case this group would have become a “monoculture with amino acid supplementation”.

Thus, we deliberately excluded the control requested by the reviewer, because the biological reality of this particular system precluded a meaningful interpretation of the corresponding results. While we agree that full factorial designs are powerful approaches to disentangle the effects stemming from multiple treatments simultaneously, they can unfortunately not be meaningfully applied to cases, in which the focal experimental parameters (here: absence/ presence of mutualism and amino acid supplementation) cannot be independently varied.

However, does the lacking “coculture with amino acid supplementation” group affect the interpretation of the results? Not having included this control treatment means that it remains unclear, whether the amino acid treatment itself affected the ability of strains to respond to the evolutionary selection pressure (i.e. antibiotic treatment). This could happen in two ways:

I. Auxotrophic strains are able to grow independently from their partner, which in turn may affect their propensity to evolve resistance to the applied antibiotics.

This possibility is sufficiently controlled for in the treatment “monocultures plus amino acids”.

II. Amino acid supplementation affects the resistance levels of strains to the antibiotic *per se*.

While in the evolution experiment, this second effect could not be clearly separated from the first one due to the reasons outlined above, we addressed this issue in a separate control experiment. Here, we determined the initial MIC and growth rates of amino acid supplemented mono- and cocultures and compared the resulting values to the ones of unsupplemented cocultures. This analysis provided the following insight:

Even though unsupplemented cocultures were generally more sensitive to the four tested antibiotics than amino acid supplemented cocultures, in two out of the four cases tested they were more resistant to antibiotic treatment (i.e. chloramphenicol, tetracycline) than one of the amino acid supplemented monocultures (i.e. the tryptophane auxotrophic mutant) (Supplemental Fig. 3C,D). This finding shows that the reduced ability of cocultured auxotrophs to adapt to increasing stress levels was not only due to an increased sensitivity to this stress at the outset of the experiment, but also stemmed from the ecological interaction between different genotypes.

To address the comment of the reviewer, we have:

- included a paragraph in the materials and method section that explains the rationale of the experimental design used (lines 445-457).

- included two new figures in the supplemental materials that report on the results of the newly performed experiment, in which the growth levels and MIC of ancestral groups were analysed (Supplemental Figs. 2, 3).

- included a new paragraph in the results section to discuss the newly included data and the implications for the interpretation of the evolution experiment (lines 105-125; Supplemental Figs. 2, 3).

The third section has a new experiment, which then compares coevolved versus monoevolved. It does this with a balanced design, looking at evolution with and without amino acids. However, again the same problem arises, because the strains aren't just coevolved versus monoevolved. Their history from the first experiment is coevolved and without amino acid, versus monoevolved and with amino acid (i.e. coevolved/monoevolved was confounded with amino acid supplementation). I feel bad pointing this out. The results make sense, and I suspect are probably 'true'. But without the appropriate control this can't be said scientifically. Control treatments are fundamental to experimental evolution – I have focused on the most basic possible, but confidence could probably have been increased even further, by trying to develop others as well.

The aim of the experiment referred to by the reviewer was to analyse how the coevolutionary history and the mutualistic dependence affects the resistance of consortia to different concentrations of antibiotics. In this case, the potential problems of the "coculture with amino acid supplementation" treatment as outlined above are less severe, because the experiment only considers one growth cycle. Because the 15 growth cycles of the evolution experiment would have amplified the shortcomings of the "coculture with amino acid supplementation" (see above), we deliberately excluded this treatment from the experimental design.

Thus, the two treatments "coculture without amino acid supplementation" and "monoculture with amino acid supplementation" were the only two treatments that resulted in clearly interpretable results. The newly conducted experiment, where the effect of amino acid supplementation on the populations' MICs was analysed also showed that on average, the two "monoculture with amino acid supplementation" treatment groups closely resembled the behaviour of the "coculture with amino acid supplementation" group (Supplemental Fig. 3). From the perspective of the individual strains, the only difference between these two treatment groups was the source of the amino acids (i.e. mutualistic partner versus environmental supplementation).

Taken together, we think the reviewer is right in pointing out the fact that a full factorial design would have been ideal. However, technical issues associated with some of the experimental groups required us to adapt the experimental design accordingly. Test experiments, performed to assess the potential difference to the now missing experimental groups showed that the results are indeed comparable and provide meaningful insights into the way mutualistic dependence affects the ability of strains to evolve.

Minor comments:

1. Introduction, first+ third paragraph. Does this also relate to Bob May's classic theory (his 73 book *Stability & Complexity in Model Ecosystems*) on how mutualism population dynamics are less stable?

Using the basic structure of Lotka-Volterra equations, May derived a set of "qualitative conditions" ((i)-(v), pg. 71), allowing for a comparison between the stability of different forms of associations, including predator-prey (+,-), commensalism/amensalism (+,0 / 0,-), competition (-,-) and mutualism (+,+). He concluded that competition and mutualism are not consistent with "qualitative stability." As measured by May, stability can be derived by the sign of the Eigenvalues of the Jacobian Matrix of the system of equations describing the interaction. According to this criterium, a system is stable when, after a perturbation inducing a change in the population size, the system returns to its initial state.

In our study, however, we do not simply change the initial population size, but instead, we change the environment where populations grow (i.e. increase concentration of antibiotics). In the context of a model, this is equivalent to simultaneously changing both the population size and the parameters affecting population growth. Moreover, in our study, the coculture collapses because of its inability to evolve resistance. In that sense, our concept of stability is closer linked to an evolutionary response than an ecological one.

Additionally, the model structure used in May's analysis is fundamentally different from the mutualistic association in our study: the Lotka-Volterra model assumes a facultative association between the mutualistic partners, while in our case the nature of the association is obligate. For these reasons, we decided to not include or mention May's model in our manuscript.

2. Line 107. I know it cuts straight from intro to results, but it would be useful to provide a bit more summary of what the actual selection experiment was, to provide context to the results (to go with fig 1 – which was very clear).

We thank the reviewer for this comment and agree that given the questions our experimental design has obviously raised, it makes sense to explain in detail the reasons that have led to this decision. In this context, also the newly performed experiments (i.e. growth and MIC of ancestral consortia) can be discussed, because they provide the baseline for the changes observed in the evolution experiment. In the revised version of the manuscript, a new paragraph has been included that addresses these issues (lines 105-125).

Reviewer #3

This is a good manuscript with valuable results. It requires few modifications, as some points need further explanations, and some aspects could be explained more briefly. Some weak points of the paper consist of the fact that the two strains used in the experiment are treated as black boxes, with no indications on the genomics or on the metabolism of the bacteria. We do not know why the strains react in the way they do.

We thank the reviewer for the positive comments on our manuscript. Below we explain how we responded to the other points raised.

The authors perform experiments using two modified strains of *E. coli* to demonstrate a number of points regarding cooperation and evolvability under environmental stress.

1. Metabolically interdependent strains grow less under antibiotic stress than strains on their own.
 2. Coevolution can enable or restrain the ability of interaction partners to respond to changing environmental selection pressures.
 3. Under stress, mutualistic interactions can collapse and strains can regain metabolic autonomy.
- Below I included observations regarding each of these points:

(1) For the first point, the authors find “surprising” that “cocultures of mutualistically interacting *E. coli* strains reached lower population densities and reduced resistance levels as compared to the corresponding monoculture controls (Figs. 2, 3, and Supplementary Fig. 1)” (Discussion, Line 265). To support this argument, the authors cite studies that demonstrate mutualistic relationships of two “unequal” species, in which resistant strains can protect the sensitive ones. In the particular setup of this study, both participating strains are more or less equal to each other (both are resistant or susceptible to the antibiotics, there is no one strain more resistant than the other, and their growth rates in absence of antibiotics is not shown, but are presumed similar to each other).

In the revised version of the manuscript, a new experiment has been included that analyses the growth rate and the susceptibility to antibiotics (MIC) of the ancestral strains/ consortia in the absence and presence of amino acid supplementation. The results show that despite the fact that the genotypes differ in just one mutation (i.e. a different amino acid biosynthesis gene has been deleted), both growth rates and MICs are significantly different from each other (lines 127-150; Supplemental Figs. 2, 3).

A simple mathematical demonstration can show that the result of this experiment is predictable.

Let A be the fitness of the first strain and B the fitness of the second strain.

Then $A * B$ = the initial fitness of the strains A and B growing in coculture. The initial total fitness of strains A and B growing separately is $A+B$.

When exposed to antibiotics, the fitness of the two strains incurs penalties A_1 and B_1 ($0 < A_1 < 1$ and $0 < B_1 < 1$), as follows:

Fitness of A becomes $(A - A \cdot A_1)$ and the fitness of B becomes $(B - B \cdot B_1)$. Thus, the new fitness of the cocultures is:

$(A - A \cdot A_1) + (B - B \cdot B_1) = A \cdot B - A \cdot B \cdot B_1 - A \cdot B \cdot A_1 + A \cdot B \cdot A_1 \cdot B_1 = A \cdot B - A \cdot A_1 \cdot (B - \frac{1}{2} \cdot B \cdot B_1) - B \cdot B_1 \cdot (A - \frac{1}{2} \cdot A \cdot A_1)$, where $A \cdot B$ is the initial fitness, before the antibiotics are applied. (1)

The new total fitness of the strains A and B growing separately is: $A + B - A \cdot A_1 - B \cdot B_1$, where $A+B$ is the initial fitness. (2)

The terms in the second and third positions in statement, showing decrease of initial fitness (1) are greater than the terms showing decrease of fitness in statement (2), as $A \cdot A_1 \cdot (B - \frac{1}{2} \cdot B \cdot B_1) > A \cdot A_1$ and $B \cdot B_1 \cdot (A - \frac{1}{2} \cdot A \cdot A_1) > B \cdot B_1$. Thus, the strains growing in coculture are expected to have lower fitness and reach lower population densities than the strains growing in isolation, when antibiotics are administered.

We thank the reviewer for this thoughtful response. We agree that in hindsight, the results of our experiments might have been predictable with a relatively simple model. However, several factors likely affected the observed outcome, which would not have been part of this simplified explanation. Examples include the rate of mutations causing cross-protection, epistatic interactions of mutations present in different genomes, or the population dynamics in different experimental groups. Given these unknowns, the simplified model would have had very little predictive power. This is why we prefer to not include it into the manuscript.

(2) This is a very neat experiment, with nice results. However, some details on genomic mechanisms that lead to AMR, in monocultures and in cocultures, should be included.

We believe that including data on the genomic mechanisms that gave rise to the observed resistance level will not add information that is relevant to the current research focus, but open up an entirely new field of questions. Given that mutations evolved in the context of an ecological interaction, it would not be sufficient to simply provide a list of mutations, but subsequent experiments are necessary to reconstruct the mutations in the ancestral genotypes and verify their phenotype in a mono- and coculture context. Given the time and effort that would have to be invested, the expected insights are unlikely to significantly enhance understanding of the current work. In contrast, including a completely new data set of this scale would complicate understanding of the already very comprehensive study. We plan to address this issue in a separate study that exclusively focusses on the evolutionary genetics of the performed evolution experiments, in which both genomic data and phenotypic analyses will be presented.

(3) This is also a very nice result. However, the two E. coli strains are described like black boxes. Why and how does the Tyrosine-auxotrophic mutant regain metabolic autonomy, but the tryptophane-auxotrophic does not? An explanation regarding the strains different metabolism or genetics should be included.

Most likely, auxotrophic strains regained prototrophic phenotypes by co-opting the function of another gene. This means that the newly evolved biochemical capability is likely less efficient in producing the required metabolite than the originally deleted gene. Despite these fitness costs, high levels of environmental stress favour a (presumably low-fitness) autonomy over mutualistic dependence. The questions how this is achieved genetically and why mutants differ in their ability to do so are very interesting, yet in our view not essential to the main focus of the current manuscript. To address the point raised by the reviewer, a statement has been included in the revised manuscript, in which potential explanations for this observation are discussed (lines 383-391).

Few more points:

(4) Line 138 mentions an unsupervised learning algorithm to identify the difference and similarities in the evolutionary trajectories of monocultures and cocultures. This is not clear. What clusters does this refer to? It is not clear what this analysis is trying to show.

During the long-term evolution experiment, strains have to adapt to an increased concentration of antibiotics. Such an adaptive process leads to a differential response in cultures, measured as the changes in cell densities (OD_{600nm}). The profile of cell density variation during the experiment defines an evolutionary path followed by mono and cocultures.

An unsupervised learning algorithm was used to characterize the evolutionary path followed by mono- and cocultures of auxotrophs. In particular, profiles of cell densities between cultures and across transfers in the long-term evolution experiment were compared using a variational Gaussian mixture algorithm to find clusters in the data. In other words, each path (i.e. a leaf in the graph) is grouped into different categories, and their similarities are depicted by the distance between the leaves.

This point has now been clarified in the revised version of the manuscript (lines 177-179, 573-576).

(5) Line 148: The authors show that cocultures of strains that adapted together grow better than cocultures of strains that evolved separately, when exposed to TET and Chloramphenicol. This trend changes, when the coculture is supplemented with required AA: the supplemented cocultures of strains that evolved separately grow better than supplemented cocultures adapted together. Were the types of cocultures (with and without AA, adapted and not adapted) compared, in the absence of antibiotics? It appears obvious that supplementing the non-adapted coculture with AA leads to better growth than the supplemented adapted one.

In response to the reviewer's comment, we have performed a new experiment, in which the growth of all experimental groups analysed in Fig. 4 was determined in the absence of amino acid supplementation (Fig. 4 C,F). In line with the reviewer's prediction, results of this new data set revealed similar trends as observed in the antibiotic-treated cultures, yet in a slightly weakened form. In the absence of antibiotic-mediated selection, coevolved strains could utilize supplemented amino acids significantly less well than cocultures of monoevolved strains (Fig. 4C,F). This observation suggests that there is a factor limiting the growth of coevolved auxotrophs, which likely stems from the physiological interdependency between both strains that has tightened during the coevolutionary process. In the revised version of the manuscript, we have included the new data set (Fig. 4) and discussed the results (lines 263-269).

(6) Regarding Figure 2: It is surprising that resistance was acquired so fast. We see little drop in population density after the sub-MIC dash line in the Tetracycline plot. Why is this?

Previous studies have showed that exposure to sub-lethal concentrations of antibiotics can, in some instances, affect the strain's ability to evolve increased resistance levels to the corresponding antibiotic. One possible mechanism is an increased mutation rate that may be due to the increased production of reactive oxygen species. Given that this has previously been shown to be the case for *Escherichia coli* populations adapting to tetracycline (Kohanski *et al.* 2010 *Molecular cell*), similar mechanisms may also have operated in our experiment.

(7) Regarding Figure 2: I suggest that instead of Fig2 E and F, confidence intervals could be added to Fig2 C and D.

The data displayed in Fig. 2C,D and Supplementary Fig. 4C,D are absolute values of surviving replicates, which do not allow to calculate any confidence intervals.

(8) Figure 3: What do the colours of the box plot mean? There is no legend. It is not clear what the letters above the box plot mean (differences between the strains).

The different colours of the boxplots correspond to the colour code we also used in the previous figures to denote the different experimental groups. We did not include a legend in the figure, since the colours are already explained by the labels of the X-axis. However, to clarify this point, we included a statement in the legend of the figure that explains the colour code (lines 907-909). We also explain in the legend that different letters above boxes indicate significant statistical differences between the different experimental groups (lines 911-913).

(9) In Fig 3A: Does this mean that supplementation of TRP leads to worse results than no supplementation at all (CO- versus TRP+)? Can this be clarified in text?

The two strains that were used to initiate the focal evolution experiment have been previously coevolved for 80 d and, as a result of this, have evolved an obligate and cooperative cross-feeding interaction. One interesting aspect of this process was that strains reduced or even lost their ability to utilize environmentally supplied amino acids. This observation may also explain why in some cases their ability to evolve resistance to antibiotics (e.g. chloramphenicol) was less dependent on supplementation with amino acids. We clarify this point in lines 206-215.

Reviewer #4 (Remarks to the Author):

The author's present a clear and accessible set of experiments examining the stability and adaptability of a two-strain community stabilized by a mutualism relative to populations founded by corresponding individual strains. I think the central results are clear and strong.

We thank the reviewer for the positive feedback.

I do, however, have one overarching concern – the key results are not accompanied by follow-up sufficient to determine even their general basis. This limits my ability to evaluate how they can be generalized.

In the following, we did our best to clarify this point.

Major comments:

(1) I very rarely critique a manuscript for what is absent (barring specific controls). Here, though, I can't help but think that a fuller explanation of the basis of at least the main finding – that cocultures are less well able to adapt than component strains – would greatly increase the impact of the work.

By way of example – a possible mechanism for reduced coculture evolvability is that coculture populations simply reach lower densities, and thus have less mutational input, than do monocultures. Population densities recorded at the end of the first and second transfer (Fig. 2AB and S1AB) – when no antibiotic was present – show a lower density for cocultures, which would be consistent with this possibility. (They also show a big decline in coculture population size at the end of the first transfer relative to the second, suggesting that they were not well adapted to the base medium.) To isolate this effect, it would be interesting to have compared evolvability with population size controlled for (perhaps by reducing the input of supplemented amino acid to monocultures) so that the influence of strain interactions could be isolated. I wonder also if the apparently lower amount of limiting amino acid in cocultures relative to monocultures influenced the results reported in Fig. 5. If coculture strains were under stronger selection for prototrophy, because environmental amino acids were not sufficient for fast growth of auxotrophs, then a higher frequency of reversion to prototrophy would be expected. Perhaps it can be argued that this kind of effect is the point of the experiment – cocultures are costly – but the fact that the amount of amino acid added to cocultures, and thus, population sizes and strength of selection for prototrophy in monocultures, is presented here as being arbitrary, for me undermines the significance, or at least the generalizability of results.

In summary, I'd really like the authors to make some effort to speak to the basis of the main results they obtain.

We thank the reviewer for these comments. While we agree that populations differed initially in their population size, we do not think that the resulting differences (e.g. in mutational supply) can explain the observed patterns. In the following we would like to explain our reasons for this interpretation:

(1) Differences in initial populations of monocultures and cocultures were only around 25%. Furthermore, over the course of the evolution experiment, in three out of the four cases analysed did population sizes of unsupplemented cocultures reach population densities that were similar or comparable to the one of one of the two monocultures (i.e. the tryptophan auxotroph) (Fig. 2A; Supplemental Fig. 4A,B). Given the total size populations achieved in the evolution experiment, we believe that the initial differences in population size

were marginal and unlikely to have limited the supply of mutations to an extent that could explain the observed results.

(2) Published studies show that in bacteria including *Escherichia coli*, mutation rates are higher in smaller populations and lower in larger populations (see e.g. Krašovec *et al.* 2017 PLoS Biology). This pattern suggests that the effect mentioned by the reviewer is unlikely to have caused the differences between experimental groups.

(3) Especially in the beginning of the experiment (i.e. transfers 2-6) when sub-lethal concentrations of antibiotics have been applied, the growth pattern of unsupplemented cocultures differed significantly from the one of amino acid-supplemented monocultures. While densities of cocultures increased significantly, monoculture densities decreased in all four cases (results of statistical test not shown). This observation shows that the growth difference between mono- and cocultures was strongly determined by effects stemming from the cooperative interaction.

(4) Strains that have been used for this study were derived from a previous evolution experiment, in which cocultures of auxotrophic bacteria have been serially propagated in the same minimal medium as used for this study for 80 d (Preussger *et al.* 2020 *Current Biology*). During this time, strains underwent significant genotypic and phenotypic alterations that coincided with a strongly increased growth under coculture conditions. Hence, it is likely to assume that these strains were phenotypically well-adapted to the minimal medium used. The reason why cocultured populations experienced a drop in their population density before they started to recover is most likely due to the fact that the obligately dependent strains need some time to establish and reach an equilibrium. This is why we started with the antibiotic treatment only after the second transfer of the experiment.

In the revised version of the manuscript, we have included a new paragraph in the discussion section, in which we elaborate on the questions raised by the reviewer (lines 336-350).

2. The MIC follow-up (Fig. 3) to the evolution experiment outcome presented in Fig. 2 is interpreted to show that differences in monoevolved and coevolved population MICs likely impacted adaptability. I'm trying to reconcile the differences in scale. If I understand correctly, the evolution experiment ended at antibiotic concentrations below the ones obtained by every group plotted in Fig 3. How should we interpret, for example, that 'resistance levels reached by coevolved consortia was significantly lower than one of the monoevolved strains' when that low resistance was nevertheless above the level required for growth in the original evolution experiment. Why should MIC above the selected value be relevant to population success? A clearer assessment of this experiment to the outcomes of the main evolution experiment would be very helpful.

The experiment presented in Fig. 3 aimed at analysing resistance levels reached by the different consortia towards the end of the evolution experiment. The premise of this is that experimental conditions that favour an increased evolvability should also result in more mutations that confer higher resistance levels. Thus, MIC is used as a proxy for evolvability. Even if populations did not experience such high antibiotic concentrations during the evolution experiment, a higher MIC still indicates that within the populations analysed, some genotypes existed that could have survived under these conditions.

In the revised version of the manuscript, we included a sentence to clarify this point (lines 316-319).

Minor comments:

Small typographical issues, most I haven't detailed. L50. Explain the mutualism relevant to angiosperms as an example of a positive consequence of a mutualism for evolvability.

In the revised version of the manuscript (lines 50-53) we have explained that the mutualism between angiosperms and their pollinators has triggered a rich adaptive radiation. In other words, this is a possible example of how a mutualistic interaction may have increased the evolvability of the species involved.

L62. 'extent' to 'extend'

L70. 'system' to 'systems'

L188. "allowed *us* to"

Corrected as suggested.

L191, Fig 4. I suggest that this analysis should account for different initial levels of adaptation to the antibiotics. My guess is that changes are conservative to this factor (e.g., in Tc monocultures start being slightly better adapted than strains isolated from cocultures, though the coculture strains go on to reach higher ODs across all tested Tc concentrations), but it would be good to be reassured.

It is not clear to us how we would normalize this data to account for initial resistance levels, since the groups analysed represent combinations of genotypes that emerged in the course of the evolution experiment. Also, the observed growth levels were a combination of the resistance levels of individual genotypes and the ecological interaction between them. Thus, it is not possible to normalize for ancestral resistance levels without making very strong assumptions that are possibly violated. To still, provide information on the resistance levels of ancestral populations and consortia, we report the results of a new experiment (Supplemental Fig. 3).

L216. Needs more context.

We have included an example to clarify which kind of costly obligate interactions we refer to in following paragraph (lines 275-278).

L258. It would be useful to clarify how this (ecological?) effect influences evolutionary potential.

We specified the point we wanted to address in this section (lines 316-322).

L270. I would have thought that cross-protection would depend on the nature of the resistance mechanism? E.g., Tc resistance might involve upregulation of efflux pumps, potentially creating a relatively higher concentration of the drug around a resistant cell. Although the actual resistance mechanisms that evolved might not be known, it is probably unlikely that they would involve modification/deactivation of the drug, which is the mechanism most likely to confer cross-resistance. Perhaps this is the point the authors are getting at when they note that resistance had to evolve de novo in this work? If so, it would be useful to clarify.

We used a brought variety of antibiotics to assess if cross-protection is playing a critical role in our experimental set-up. Especially a resistance to ampicillin, which is typically facilitated by beta-lactamases that degrade the antibiotic, was a candidate where we expected to find cross-protection. However, we did not find any evidence for such an effect. Given that the exchange of amino acids depended on a close physical contact between cells (Preussger *et al.* 2020 *Current Biology*), the evolution of a mechanism that confers cross-protection would have protected both the mutant and other, sensitive cells. Thus, mutualistic cross-feeding would have amplified the effect of cross-protection. Since this was not observed, we conclude that cross-protection did not seem to play a role in our experiment. We have now reworded the corresponding paragraph to clarify this (lines 351-358).

Fig. 2. I think it is up to the authors, but I would suggest clarifying that the X-axis here indicates increasing antibiotic concentrations. That information is clearly in the text and legend, but interpretability of the top four panels could be helped with inset of an 'Ab concentration 'triangle' or similar.

We added the suggested antibiotic concentration triangle to enhance interpretability of these graphs.

Fig. 2 EF. To me this analysis doesn't add much – it is clear from panels A-D that cocultures are less adaptable (including associated statistical analysis). My suggestion is to move to the supplementary material.

These figures represent a graphical illustration of a statistical comparison between the evolutionary trajectories of experimental groups. Even though the growth data of panels of Fig. 2A,B and Supplemental

Fig. 4A,B have been used for this, the result is not immediately obvious, yet crucial to support our argument. To facilitate readability of our manuscript, we prefer to keep the figures in the main text, rather than moving them to the supplement.

To clarify what exactly this analysis does, we now have included an explanation in the main text (lines 177-179, 573-576).

Reviewers' Comments:

Reviewer #1:

Remarks to the Author:

No further comments to add. The authors have expertly addressed all my concerns.

Reviewer #2:

Remarks to the Author:

The authors have made a very thoughtful and positive effort to take on the feedback from the referees, and adjust their paper accordingly.

My earlier review was very positive about the question and writing, but worried about the controls in the experimental design. In particular, that "there appears not to be a treatment where the two strains were co-cultured, and the required amino acid was supplied." This would matter because it would mean that the consequence of amino acid level and whether strains were co-cultured could not be separated.

I am a bit mixed about the authors response to this worry, but they have convinced me.

The co-culture with amino acid treatment was not carried out.

(1) The authors state that this would have been based on the expectation that it completely eliminated mutualism. However, this would not have been required – it is just separating the two experimental factors, amino acid and co-culture.

(2) The authors state that strain frequencies would have varied. However, if that is the result, that is the result. The comparison is the influence of different experimental factors- amino acids and mono-/co-culture.

Points 1 + 2 didn't convince me, and I wonder if we are talking at cross purposes – the authors focus on "the only two treatments that resulted in clearly interpretable results", whereas I am talking about balancing experimental treatments even if they are a bit messy, so as to be able to separate the influence of those treatments.

(3) The authors focus on comparing mutualism versus non-mutualism as the treatment (which can be done while acknowledging that this involves manipulating multiple factors).

(4) Supporting analyses on MIC and amino acid levels.

Points 3 + 4 seem much more useful to me. The experimental design can be justified by 3, and 4 is some nice additional data to deal with a potential issue. I still think it would have been better to carry out the + amino acid + co-culture control – it is just a control, the results can be messy. But it is possible to get away without it.

The authors have added in a clear and balanced discussion of the experimental design. I think they could also be a bit more explicit with how the design is two factors to make "mutualism without amino acids" versus "non-mutualism with amino acids", but that could be done quite easily.

Reviewer #3:

Remarks to the Author:

All my comments are based on the Document with tracked changes:

[https://mts-](https://mts-ncomms.nature.com/ncomms_files/2021/05/18/00284782/01/284782_1_rebuttal_5580595_qtjzys_convrt.pdf)

[ncomms.nature.com/ncomms_files/2021/05/18/00284782/01/284782_1_rebuttal_5580595_qtjzys_convrt.pdf](https://mts-ncomms.nature.com/ncomms_files/2021/05/18/00284782/01/284782_1_rebuttal_5580595_qtjzys_convrt.pdf)

I thank the authors for presenting their new experiments (as described starting with line 130) and changing the Results and Discussion sections accordingly.

Although the arguments presented in the Discussion (lines between 345 and 359) are good, I

believe that a new figure is needed, to present differences in growth rate and MIC of evolved monocultures and unsupplemented coculture, adjusted with respective values of ancestral monocultures and unsupplemented coculture (which can serve as a baseline). The current figures only allow us to see the differences between the evolved monocultures versus evolved co-culture, however, to support the points presented in the Discussion (lines between 345 and 359), the values would need to be normalized.

Other observations: 1) Supplementary figures do not have legends. 2) Authors argue at line 117 that they did not grow the coculture with supplementation of amino acids, because this leads to competition between strains. However, on several occasions, they do mention experiments with coculture in the presence of amino acids. (eg. line 138, line 142, Fig 3 and all supplementary figures with CO+)

Reviewer #4:

None

REVIEWER COMMENTS

Reviewer #1:

No further comments to add. The authors have expertly addressed all my concerns.

We thank the reviewer for the time and effort to evaluate our revised manuscript.

Reviewer #2:

The authors have made a very thoughtful and positive effort to take on the feedback from the referees, and adjust their paper accordingly.

My earlier review was very positive about the question and writing, but worried about the controls in the experimental design. In particular, that “there appears not to be a treatment where the two strains were co-cultured, and the required amino acid was supplied.” This would matter because it would mean that the consequence of amino acid level and whether strains were co-cultured could not be separated.

I am a bit mixed about the authors response to this worry, but they have convinced me.

The co-culture with amino acid treatment was not carried out.

(1) The authors state that this would have been based on the expectation that it completely eliminated mutualism. However, this would not have been required – it is just separating the two experimental factors, amino acid and co-culture.

(2) The authors state that strain frequencies would have varied. However, if that is the result, that is the result. The comparison is the influence of different experimental factors- amino acids and mono-/co-culture.

Points 1 + 2 didn't convince me, and I wonder if we are talking at cross purposes – the authors focus on “the only two treatments that resulted in clearly interpretable results”, whereas I am talking about balancing experimental treatments even if they are a bit messy, so as to be able to separate the influence of those treatments.

(3) The authors focus on comparing mutualism versus non-mutualism as the treatment (which can be done while acknowledging that this involves manipulating multiple factors).

(4) Supporting analyses on MIC and amino acid levels.

Points 3 + 4 seem much more useful to me. The experimental design can be justified by 3, and 4 is some nice additional data to deal with a potential issue. I still think it would have been better to carry out the + amino acid + co-culture control – it is just a control, the results can be messy. But it is possible to get away without it.

The authors have added in a clear and balanced discussion of the experimental design. I think they could also be a bit more explicit with how the design is two

factors to make “mutualism without amino acids” versus “non-mutualism with amino acids”, but that could be done quite easily.

We thank the reviewer for this thoughtful and detailed feedback. In response to this comment, we have clarified more explicitly that we are comparing “mutualism without amino acids” with “non-mutualism with amino acids” (lines 101-116).

Reviewer #3:

All my comments are based on the Document with tracked changes:

<https://mts->

[ncomms.nature.com/ncomms_files/2021/05/18/00284782/01/284782_1_rebuttal_5580595_qtjzys_convrt.pdf](https://mts-ncomms.nature.com/ncomms_files/2021/05/18/00284782/01/284782_1_rebuttal_5580595_qtjzys_convrt.pdf)

I thank the authors for presenting their new experiments (as described starting with line 130) and changing the Results and Discussion sections accordingly. Although the arguments presented in the Discussion (lines between 345 and 359) are good, I believe that a new figure is needed, to present differences in growth rate and MIC of evolved monocultures and unsupplemented coculture, adjusted with respective values of ancestral monocultures and unsupplemented coculture (which can serve as a baseline). The current figures only allow us to see the differences between the evolved monocultures versus evolved co-culture, however, to support the points presented in the Discussion (lines between 345 and 359), the values would need to be normalized.

In the discussion section mentioned by the reviewer, we provide two main arguments to rule out that the observed differences in the ability of both groups to respond to the antibiotics treatment was due to the initial growth levels that differed between mono- and cocultures. These arguments are:

(1) Differences in population sizes throughout the evolution experiments were marginal.

(2) The mutation rate of *E. coli* is density-dependent and higher in smaller populations.

To support these arguments, we provide four graphs that show the change in the population density of all three treatment groups over time (Fig. 2A,B and Supplementary Fig. 4A,B) as well as a statistical test that compares differences between groups (lines 169-175). The test compares all time points and is thus the best possible way to support the two abovementioned arguments.

The reviewer suggests to graphically display how the growth rate changes over time by relating the final value to the one of the ancestral consortium. However, since this analysis would only consider a single time point, the result would be much less informative than the data and analyses already presented. In addition, the two arguments above focus on differences in population size, which can be achieved by both a high and a low growth rate. Thus, we feel that the requested figure showing

changes in relative growth rates would not help to support the above arguments. Given that the current version of the manuscript is already reporting a lot of complex data, we prefer to not include additional graphs that are not essential to the main focus of the paper.

However, we replaced the previous **Figure 3** with a new version, that shows the change in the MIC of experimental populations over the course of the evolution experiment. As requested, we normalized the MIC values relative to the values of the ancestral consortia in these new graphs.

Other observations:

1) Supplementary figures do not have legends.

In our previous submission, we uploaded two files “Supplementary Material” and “Supplementary Figures”. The file “Supplementary Material” contained all supplementary information including figures and legends, while the file “Supplementary Figures” only contained the high quality versions of the figures without providing legends. We apologize for any inconvenience this might have caused.

In the submission of our revised manuscript, we only included one file called “Supplementary Information” that contains all necessary information.

2) Authors argue at line 117 that they did not grow the coculture with supplementation of amino acids, because this leads to competition between strains. However, on several occasions, they do mention experiments with coculture in the presence of amino acids. (eg. line 138, line 142, Fig 3 and all supplementary figures with CO+)

The reviewer is right in pointing out this seeming discrepancy. The reason for why we did not include a “coculture with amino acid supplementation” treatment in the evolution experiment was that competition between both partners would likely have resulted in a loss of one of the two partners (**Supplementary Figure 1**). This effect would only manifest on the temporal scale of several transfers. However, the experiments referred to by the reviewer did not involve any transfers, but only analysed the focal parameter during a single growth period. Given that the effect of competition is likely to be negligible on this short time scale and that the amino acid supplementation treatment provided some additional insights (e.g. Did the ability of cells to utilize environmentally available amino acids change over time?), we included the “coculture with amino acid supplementation” treatment in some short-term experiments, yet not the main evolution experiment.

Reviewers' Comments:

Reviewer #2:

Remarks to the Author:

The authors have again made a thoughtful and positive response to my comments and that of the other referees. The experimental design, and justification for, are now super clear. I have no further comments - it is a great paper!

Reviewer #3:

Remarks to the Author:

I thank the authors for carefully addressing all my previous comments. I am now happy with the shape of the manuscript.

Reviewer #4:

None